# Assessment of community efforts to advance network-based prediction of protein–protein interactions

Xu-Wen Wang [1], Lorenzo Madeddu[2], Kerstin Spirohn [3,4,5], Leonardo Martini[6], Adriano Fazzone [7], Luca Becchetti[6], Thomas P. Wytock [8], István A. Kovács[8,9], Olivér M. Balogh [10], Bettina Benczik[10,11], Mátyás Pétervári[10], Bence Ágg [10,11], Péter Ferdinandy[10,11], Loan Vulliard [12,13], Jörg Menche [12,13,14], Stefania Colonnese[15], Manuela Petti [6], Gaetano Scarano [15], Francesca Cuomo[15], Tong Hao [3,4,5], Florent Laval [3,4,5,16,17,18], Luc Willems[16,18], Jean-Claude Twizere [17,18], Marc Vidal[3,4], Michael A. Calderwood [3,4,5], Enrico Petrillo [19,20], Albert-László Barabási[19,21,22], Edwin K. Silverman[1], Joseph Loscalzo [19], Paola Velardi[2] ✉ & Yang-Yu Liu [1,23] ✉

Comprehensive understanding of the human protein-protein interaction (PPI) network, aka the human interactome, can provide important insights into the molecular mechanisms of complex biological processes and diseases. Despite the remarkable experimental efforts undertaken to date to determine the structure of the human interactome, many PPIs remain unmapped. Computational approaches, especially network-based methods, can facilitate the identification of previously uncharacterized PPIs. Many such methods have been proposed. Yet, a systematic evaluation of existing network-based methods in predicting PPIs is still lacking. Here, we report community efforts initiated by the International Network Medicine Consortium to benchmark the ability of 26 representative network-based methods to predict PPIs across six different interactomes of four different organisms: *A. thaliana*, *C. elegans*, *S. cerevisiae*, and *H. sapiens*. Through extensive computational and experimental validations, we found that advanced similarity-based methods, which leverage the underlying network characteristics of PPIs, show superior performance over other general link prediction methods in the interactomes we considered.

A comprehensive understanding of the human PPI network (also known as the human interactome) could offer global insights into cellular organization, genome function, and genotype–phenotype relationships[1,2]. The discovery of previously uncharacterized PPIs could facilitate important interventional goals, such as drug target identification and therapeutic design[3]. Despite remarkable experimental efforts in high-throughput mapping, the human interactome map remains sparse and incomplete[2,4], and is subject to noise and investigative biases[2]. These factors represent a severe limitation to the accurate understanding of cellular organization and genome function. Computational methods can accelerate knowledge acquisition in biomedical networks by significantly reducing the number of alternatives to be confirmed in bench experiments[5–9]. Yet, high incompleteness of the human interactome map may reduce the effectiveness of state-of-the-art computational methods. In this context, the computational prediction of previously uncharacterized PPIs based on experimentally observed PPIs becomes a particularly challenging but potentially highly rewarding task.

**Fig. 1 | Workflow of the INMC PPI prediction project.** 26 representative network-based methods were systematically evaluated to predict PPIs in the interactome of four different organisms: *A. thaliana*[19], *C. elegans*[20], *S. cerevisiae*[21], *H. sapiens*: HuRI[4], STRING[22] and BioGRID[23] (using rTRM package[82]). During the computational validation, the PPIs of each interactome were divided into training set and validation set through 10-fold cross-validation. The performance of each method was evaluated using four standard metrics: AUROC, AUPRC, P@500, NDCG. For each method, an overall score was defined as the sum of z-scores of three metrics (AUPRC, P@500

and NDCG) for each interactome. Top-seven methods were selected based on their performance in predicting human PPIs during the computational validation. Using the entire human interactome, each of the top-seven methods predicted the top-500 human PPIs for experimental validation using the Y2H assay. PPI: protein–protein interaction. AUROC: Area Under the Receiver Operating Characteristic curve. AUPRC: Area Under the Precision-Recall Curve. P@500: Precision of the top-500 predicted PPIs. NDCG: Normalized Discounted Cumulative Gain. Y2H: yeast two-hybrid assay. v1-v3: assay 1-assay 3.

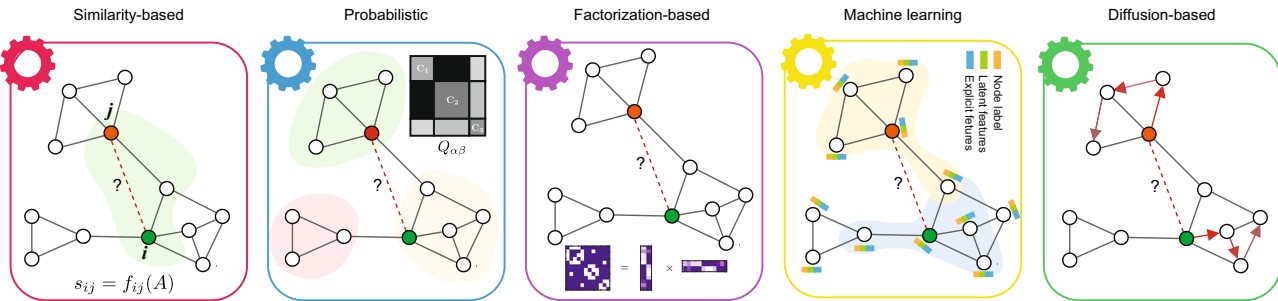

**Fig. 2 | Diagram of the five major categories of link prediction methods.** (1) Similarity-based methods. These methods quantify the likelihood of links based on predefined similarity functions among nodes in the graph, i.e., the common neighbors (green area). (2) Probabilistic methods. These methods assume that real networks have some structure, e.g., community structure. The goal of these algorithms is to select model parameters that can maximize the likelihood of the observed structure. The connecting probability of nodes within a community is higher than that between different communities (gray matrix). (3) Factorization-based: The goal of these methods is to learn a lower dimensional representation for each node in the graph by preserving the global network patterns. Next, the compressed representation is leveraged to predict unobserved PPIs by either calculating a similarity function or training a classifier. (4) Machine learning: There are numerous methods among machine learning categories; here, we illustrate this category using the state-of-the-art graph neural networks (GNN). Those methods embed node information by aggregating the node features, link features and graph structure using a neural network and passing the information through links in the graph. Thereafter, the learned representations are used to train a supervised model to predict the missing links. (5) Diffusion-based: These methods use techniques based on the analysis of the information gleaned from the movement of a random walker diffusion over the network (paths indicated by red arrows).

Recognizing the advantages and limitations of different computational methods in the context of PPI prediction is critical, providing the insight required to select the best predictive strategy[10–14]. To accelerate this process, the International Network Medicine Consortium (INMC) initiated a project to systematically benchmark 26 representative network-based methods in PPI prediction through standardized performance metrics and standardized, unbiased interactome analysis (see Fig. 1 for a summary of the workflow of this project). The 26 methods (with acronyms and brief descriptions listed in Table 1) were selected from the rich literature on link prediction[10,11,15] as well as the recent advancement in the field of deep graph learning[16,17]. Our selection covers major categories of link prediction from similarity-based methods to probabilistic methods, factorization-based methods, diffusion-based methods, and machine learning-based methods (Fig. 2). Note that three of 26 network-based methods also utilize biological data (i.e., the sequence information of proteins) in addition to the topology information for the PPI predictions.

To evaluate the performance of those methods, we need reliable and unbiased benchmark interactomes. Literature-curated interactomes of PPIs with multiple lines of supporting evidence might be highly reliable, but they are largely influenced by selection biases[2,18]. Therefore, here we focused on interactomes emerging from systematic screens that lack selection biases. For simplicity, we mainly focused on binary datasets where co-complex membership annotations are not included. We used the following six benchmark interactomes for performance evaluation: (1) A plant interactome including 2774 proteins and 6205 PPIs, derived from the PPIs in the *A. thaliana* Interactome, version 1 (AI-1) and literature databases[19]; (2) a worm interactome including 2528 proteins and 3864 PPIs, derived from *C. elegans* version 8 (WI8), which is assembled from high-quality yeast two-hybrid (Y2H) PPIs[20]; (3) a yeast interactome of *S. cerevisiae* including 2018 proteins and 2930 PPIs, derived from the union of CCSB-YI1, Ito-core and Uetz-screen datasets[21]; (4) a human interactome including 8274 proteins and 52,548 PPIs, derived from HuRI[4], which is assembled from binary protein interactions from three separate high-quality Y2H screens. To confirm the generality of our evaluations, we also used two additional interactomes derived from different methods: (5) a human interactome including 6926 proteins and 41,948 physical PPIs, derived from STRING[22] after filtering PPIs with normalized score lower than 0.9 to keep the high-confidence PPIs and (6) a human interactome including 19,665 proteins and 713,793 physical PPIs, derived from BioGRID[23].

For each of the six interactomes, we first performed 10-fold cross-validation to evaluate four performance metrics of the 26 different methods (here we refer to this process as "computational validation"). This analysis allowed us to rank the methods. Next, the top-seven methods were selected based on their performance in predicting interactions in the human interactome, and their top-500 predicted human PPIs (yielding a cumulative of 3276 PPIs) were chosen for a systematic and unbiased experimental validation through Y2H assays. In total, we validated 1177 previously uncharacterized PPIs involving 633 human proteins. To the best of our knowledge, no other consortium-based evaluations of PPI prediction algorithms in the literature incorporated such a large experimental validation effort.

In this paper, we report the results of this community effort, where we evaluated the performance of various link prediction algorithms in the context of PPI prediction and provided insights into the optimal computational tools required to detect the unmapped PPIs. We found that advanced similarity-based methods, which leverage the underlying characteristics of PPIs, show superior performance over other link prediction methods in both computational and experimental validations in the interactomes we considered. We described the details of these methods. The full datasets (including all of the benchmark interactomes and the experimental validation results), as well as the code for all of the tested methods and scoring functions, are freely provided to the scientific community.

## Results

### Correlations between different performance metrics

Figure 3 summarizes the results of the computational evaluation of all tested methods using 10-fold cross-validation. The metrics used to quantify the performance of each method are (i) AUROC: Area Under the Receiver Operating Characteristic; (ii) AUPRC: Area Under the Precision-Recall Curve; (iii) NDCG: Normalized Discounted Cumulative Gain; and (iv) P@500: Proportion of Positive PPIs, i.e., precision, in the top-500 prediction. To better rank the overall performance of different methods, we also computed a combined z-score for each method that summarizes its performance using different evaluation metrics. We next highlight the following key observations and insights from the analysis.

Considering that the distribution of links is highly imbalanced in the PPI prediction problem due to the sparsity of interactome maps across organisms[24,25], AUROC may overestimate the performance of a link prediction method, while AUPRC can provide more pertinent evaluation[26,27]. Indeed, by systematically comparing the performance

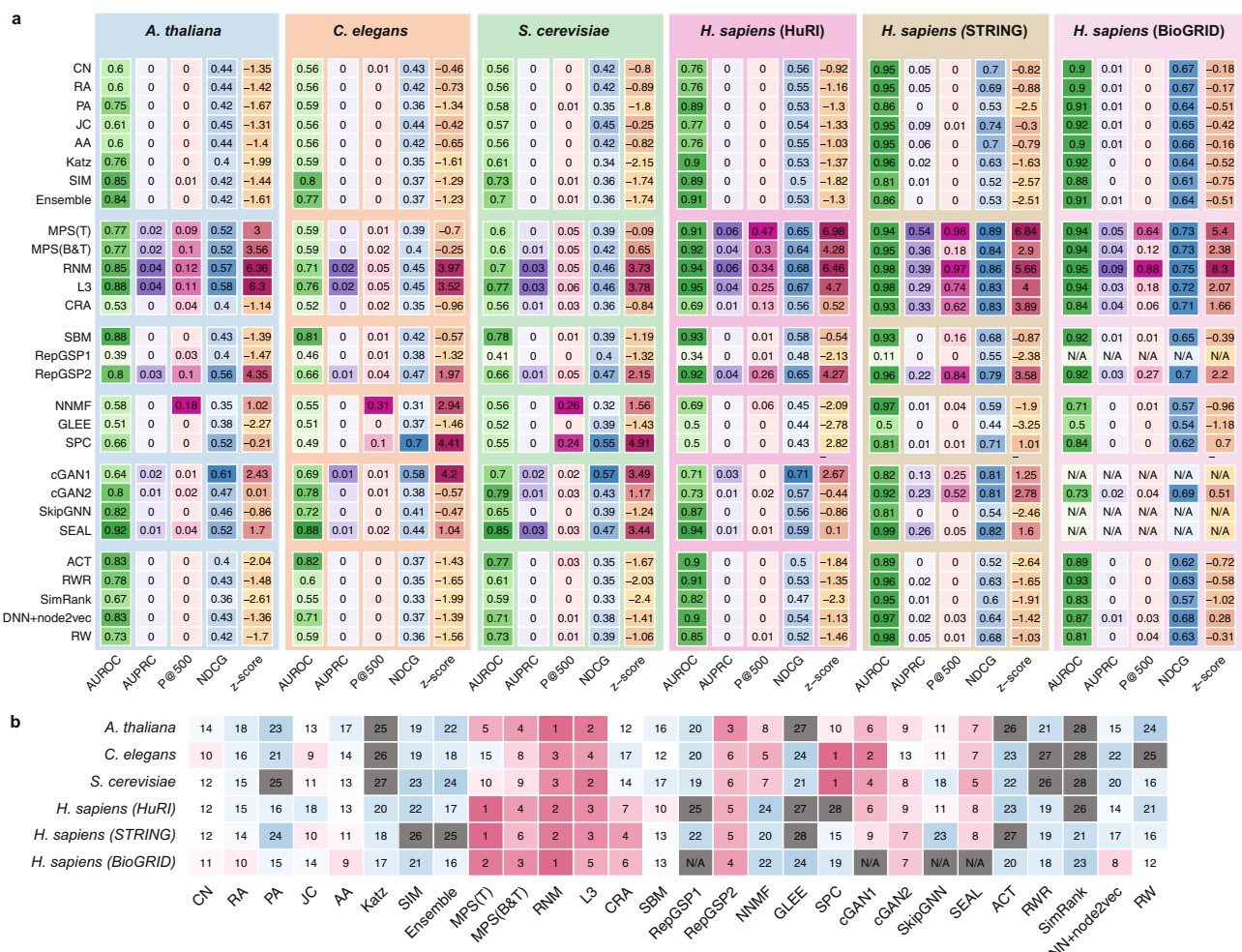

**Fig. 3 | Computational evaluation of the PPI prediction methods.** The details of each method are summarized in Table 1. **a** Heatmap plots show the performance of each method on each interactome with the following evaluation metrics: AUROC, AUPRC, P@500, and NDCG. The overall performance is calculated from z-scores of three metrics. For each metric, darker color represents better performance. **b** The ranking of the 26 methods on the six interactomes by z-scores. Note that, the performances of ReGSP1, cGAN1, SEAL and SkipGNN on the BioGRID database were not evaluated due to the prohibitive computational cost. We marked their rankings as N/A. Note that AUROC was excluded in calculating the combined z-score and ranking for each method.

metrics of various PPI prediction methods, we found clear evidence that AUROC largely overestimates the performance of any particular method. For example, the average AUROC over 10-fold cross-validation of one of the top methods, SEAL, in *H. sapiens* (HuRI) is 0.94. This very high AUROC might lead us to mistakenly conclude that SEAL is an almost perfect prediction method since the maximum value of AUROC is 1. In fact, however, we found the average AUPRC of SEAL is only 0.012, implying a poor performance in finding PPIs (Fig. 3a). Since AUROC has been widely used in the link prediction literature[28–32], we investigated whether or not the AUROC-based ranking of link prediction methods is consistent with rankings based on AUPRC or other metrics. First, we calculated the correlations between AUROC and other metrics over all methods (see Supplementary Fig. 1a–c). We found that AUROC is significantly correlated with AUPRC (Spearman $R = 0.75, p < 2.2 \times 10^{-16}$) and NDCG (Spearman $R = 0.76, p = 1.6 \times 10^{-14}$), but not with P@500 (Spearman $R = 0.18, p = 0.028$). Second, we found that the combined z-scores of different methods, including and excluding the AUROC metric, are quite consistent (Pearson $R = 0.97, p < 2.2 \times 10^{-16}$, see Supplementary Fig. 1d). Note that the strong correlation between AUROC and AUPRC does not mean that the former is as good as the latter in evaluating the performance of any particular link prediction method. Instead, it just means we can still use AUROC to rank different methods (and the ranking will be roughly the

same as if we use AUPRC), even though the AUROC values are systematically inflated (due to the data imbalance issue). Hereafter, we will, therefore, exclude AUROC in calculating the combined z-score as well as interpretation of the performance for each method.

**Predictability of interactomes is weak**

Notably, both AUPRC and P@500 of most methods are quite small for five interactomes, except *H. sapiens* (STRING) (Fig. 3a). This observation suggests that successfully predicting missing links among a large unmapped PPI space remains a challenging task. To quantify the predictability of each interactome, we calculated its structural consistency index $\sigma_c$ based on the first-order perturbation of the interactome's adjacency matrix[33]. It is important to note that a network is highly predictable (with high $\sigma_c$) if the removal or addition of a set of randomly selected links does not significantly change the network's structural features (characterized by the eigenvectors of its adjacency matrix). We found that *H. sapiens* (STRING) is most predictable ($\sigma_c > 0.58$) and $\sigma_c < 0.25$ for all other five interactomes (see Supplementary Fig. 2a), which is much lower than that of social networks[33] (e.g., Jazz[34] and NetSci[35] for which we have $\sigma_c = 0.65$ and 0.60, respectively). This might imply that *H. sapiens* (STRING) is the most unbiased and such a low structural consistency for other interactomes indicates that the unobserved parts of the five interactomes

considered in this project do not have very similar structural features as their currently observed part, which might be due to the high incompleteness of those interactome maps. Indeed, for the social networks Jazz and NetSci, if we remove 90% of its links, their $\sigma_c$ values drop from 0.65 to $0.16 \pm 0.06$ and from 0.60 to $0.25 \pm 0.09$, respectively (calculated from 50 random removals). We also examined the predictability of interactomes with different edge densities generated by the duplication-mutation-complementation model[36], finding that the predictability increases with edge density (see Supplementary Fig. 2b). Note that the seeming "inconsistency" between the predictability and P@500 is because predictability is essentially P@10%L, where L is the number of total PPIs. For HuRI, this is about P@5255, which is much lower than P@500 (see Supplementary Fig. 3).

## Performances of most PPI prediction methods vary considerably across different interactomes

We found that PPIs from *H. sapiens* (HuRI and STRING) were predicted more accurately than PPIs from the other three organisms despite their very similar edge density (Supplementary Table 1). To better demonstrate the performance variability of the different methods, we ranked those methods based on their combined z-scores of AUPRC, P@500 and NDCG. As shown in Fig. 3b, some methods (e.g., RNM) show quite robust rankings (or consistent performance) across different interactomes, while the rankings of other methods show very large variability across interactomes. This variability is likely due, in part, to the different network characteristics, i.e., the number of links and average degree (Supplementary Table 1). These, in turn, to some degree depend on the degree of completeness of existing PPIs, which can vary broadly across different interactomes[37]. To evaluate the mean performance versus variability systematically, we calculated the mean and standard deviation of rankings across all six interactomes for each method, respectively, finding that RNM yielded the highest mean ranking and lowest variability. This suggests that RNM is robust and able to perform well on interactomes with quite different network characteristics (Supplementary Fig. 4).

## Traditional similarity-based methods do not perform well

Traditional similarity-based link prediction methods have been heavily used in benchmark studies[10,11]. These methods are based on a range of simple node similarity scores, such as the number of common neighbors, the Katz index, the Jaccard index, and the resource allocation index, to quantify the likelihood of potential links (see Table 1). Hence, they are more interpretable and scalable than other methods. However, the predefined local and global similarity score functions might significantly impact their predictive power for some networks, such as PPI networks. For example, the mere presence of two proteins in an interactome with many common neighbors does not necessarily imply that they should be connected to each other because interacting proteins are not necessarily similar and similar proteins do not necessarily interact[38]. We found that most of the traditional similarity-based methods, i.e., common neighbors (CN), Adamic-Adar index (AA), and Jaccard index (JC), yield negative z-scores, indicating their performance is below the average performance of all methods (Fig. 3a).

## There are six consistently high-performing methods

Among all the methods we tested, we found that RNM, L3, MPS(T), MPS(B&T), RepGSP2 and SEAL yield relatively high AUPRC and P@500 in their computational evaluations over six interactomes. RNM is an advanced similarity-based method that integrates a diagonal noise model, a spectral noise model, and the L3 principle which shows that two proteins are expected to interact if they are linked by multiple paths of length three in the interactome[38]. RNM displays excellent performance across all six interactomes: it ranked No.1 in the PPI prediction for *A. thaliana* and *H. sapiens* (BioGRID) interactomes, and

No.2 for *C. elegans, S. cerevisiae, H. sapiens* HuRI and STRING interactomes. MPS(T) and MPS(B&T) are also advanced similarity-based methods. MPS(T) leverages the L3 principle, but at a higher level. In particular, it highly ranks a protein pair $(i,j)$ if protein-$i$ has similar neighborhood as that of protein-$j$'s neighbors. Compared with MPS(T), MPS(B&T) also takes into account the sequence similarity of a protein pair. Both MPS(T) and MPS(B&T) display good performance in PPI prediction. In particular, MPS(T) is ranked as No.1 in the PPI prediction for *H. sapiens* (HuRI and STRING). RepGSP leverages Graph Signal Processing to learn the graph edges. SEAL is a deep graph learning model based on Graph Neural Networks (GNNs). SEAL can leverage the sequence information of proteins. Moreover, it can learn a function that maps the subgraph patterns to link existence from a given network instead of using any traditional predefined similarity indices[16]. RNM, RepGSP and SEAL show high performance consistently in the six benchmark interactomes. In particular, we found RNM and MPS(T) also show high AP@K (average precision at K) against different K values (see Supplementary Fig. 5).

It Is interesting to note that, compared with MPS(T), SEAL yields higher performance in the *C. elegans* and *S. cerevisiae* interactomes, but lower performance in the *H. sapiens* interactomes. This performance difference might be due to the different graph mining techniques used by these methods. One of the similarity metrics in MPS(T) is defined as a function of the Jaccard index of the neighborhoods of two nodes, and it is reasonable that this metric should benefit from higher degree nodes for classification rather than those with small neighborhoods. Instead, SEAL uses GNNs to define the features of a node by aggregating values from its nearest neighbors. In a case where nodes have many neighbors with similar features (e.g., nodes in the same biological modules), aggregation functions (e.g., the average function) could lead to nodes with similar embeddings, which are harder to distinguish.

## Stacking models do not perform significantly better than individual methods within each interactome

It has been suggested previously that constructing a series of "stacked" models and combining them into a single predictive algorithm can achieve optimal or nearly optimal accuracy[39]. To confirm whether a stacking model is superior to individual methods in PPI prediction, we constructed four different stacking models. Stacking-model-1 (Supervised): stack 36 individual topological predictors, which come in three types, global (functions of the entire network), pairwise (function of the joint topological properties of node pair $i,j$) and node-based (functions of the independent topological properties of node $i$ and $j$), into a single algorithm, then train a classifier to predict the missing links. Stacking-model-2 (Unsupervised): for each link, take the average of its ranking (in percentile form) calculated from RNM and MPS(T). Stacking-model-3 (Unsupervised): for each link, take the maximum of its ranking calculated from RNM and MPS(T). Stacking-model-4 (Unsupervised): for each link, we aggregated the rankings from RNM and MPS(T) using the Kemeny consensus[40] approximated by the Dowdall[41] or cRank[42] method. Interestingly, we found that none of these stacking models can significantly outperform individual methods (see Supplementary Fig. 6). In general, the advantage of Stacking-model-1 is that the meta-classifier can learn to select the best predictors through supervised training; thus, the overall performance of the stacking model outperforms any individual predictor. However, in our case, the predictors of the two highest ranking methods can be directly used in PPI predictions without training a classifier. Moreover, the overlap between the PPIs predicted by different methods is very low (see Supplementary Fig. 7 for Cohen's kappa coefficient and the Jaccard index). Therefore, simply averaging the scores (Stacking-model-2) from different methods will decrease the ranking of those correctly predicted PPIs and, hence, degrade the predictive performance accordingly. Stacking-model-3 can yield slightly higher AUROC

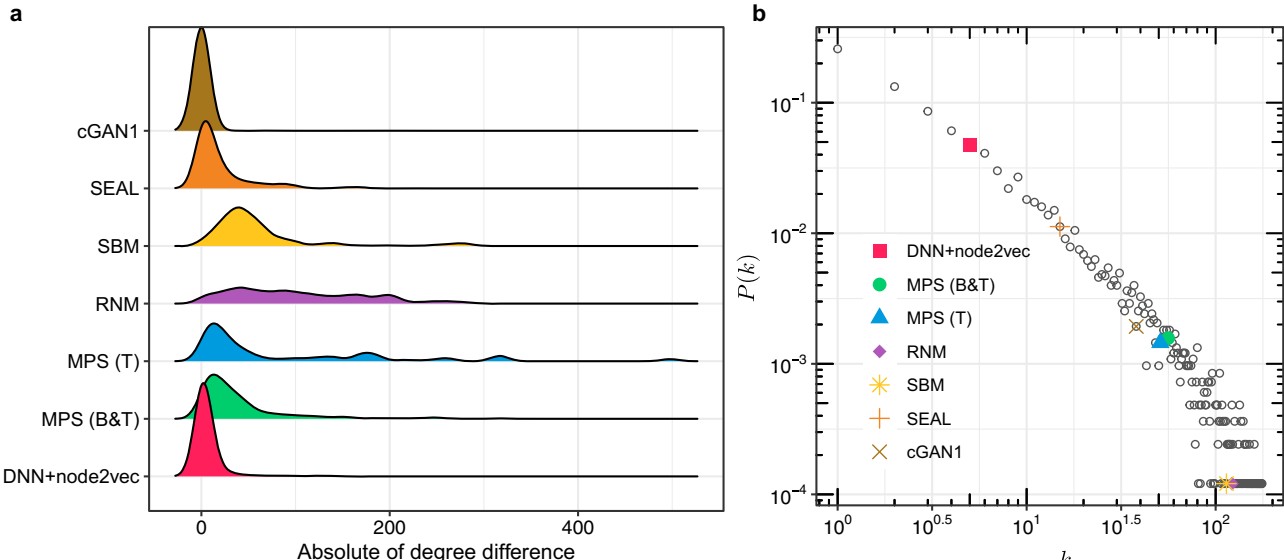

**Fig. 4 | Patterns of top-500 PPIs predicted by the top-seven human PPI prediction methods. a** For these top-seven methods, we examined the distribution of absolute value between the degrees of each protein pair. **b** Degree distribution of the *H. sapiens* (HuRI) interactome and the mean degree of proteins involved in the top-500 predicted PPIs of each method in log-log plot. *k* denotes the degree of a protein.

and NDCG than each individual method in some interactomes, but its AUPRC and P@500 still cannot surpass the best individual method. These results suggest that for a given network domain (e.g., PPI networks) stacking models do not always significantly outperform the best individual methods in link prediction.

## Patterns of predicted PPIs

Based on their performance in predicting PPIs for the *H. sapiens* (HuRI) interactome, we selected the top-seven methods: MPS(T), RNM, MPS(B&T), cGAN, SEAL, SBM and DNN + node2vec (see Methods for the selection process). To examine whether there is any particular pattern among the top-500 PPIs predicted by each of the top-seven methods, we calculated the distribution of degree difference of proteins involved in the predicted PPIs (Fig. 4a). We found that RNM, MPS(T), MPS(B&T), and SBM tend to predict PPIs involving proteins with larger degree difference than that of randomly selected PPIs in HuRI (with degree difference $16.56 \pm 1.33$ calculated from 10 random samplings of 500 PPIs in HuRI). The remaining methods cGAN, SEAL and DNN + node2vec tend to predict PPIs between nodes with similar degrees. In addition, for each method, we plotted the average degree of the proteins involved in their top-500 predicted PPIs on top of the degree distribution $P(k)$ of the human interactome (with average degree 12.7) (Fig. 4b). We found that RNM, SBM, MPS(T), and MPS(B&T) tend to predict PPIs involving proteins of high degrees, while the average degree of proteins in the top-500 PPIs from deep learning methods, such as DNN + node2vec and SEAL, is much lower. This difference could be due to the fact that RNM provides more predictions for high-degree nodes (see Fig. 4b) and MPS also leverages the L3 principle at a high level, while DNN + node2vec focuses more on local network topological structure rather than on degree. The performance of all those four methods will decrease on degree-preserving randomized interactomes (see Supplementary Fig. 8).

## Performance of prediction methods in experimental validation

To the best of our knowledge, this is an unprecedentedly large-scale experimental validation of network-based PPI prediction methods in a systematic benchmark study. To validate the performance of the PPI prediction methods experimentally, we applied each of the top-seven methods (MPS(T), RNM, MPS(B&T), cGAN, SEAL, SBM and DNN + node2vec) to the human interactome (HuRI) and predicted the top-

500 unmapped human PPIs. The union of the top-500 predicted human PPIs from the top-seven methods includes 3276 unique protein pairs. Next, we validated those protein pairs using the three orthogonal Y2H assays formerly used to obtain the HuRI map. The set of predicted PPIs were collectively recovered at a rate that was on par with the recovery of high-confidence binary PPIs from the literature (see Supplementary Fig. 9). In total, from the successfully tested pairs, we identified 1177 previously uncharacterized human PPIs (involving 633 proteins) by considering a protein pair to be positive if it is positive in at least one of the three assays, and negative if it is scored negative in all the three assays for which it was successfully tested. Note that some protein pairs were not successfully tested due to technical issues (see Methods). Overall, we found that MPS(B&T) is the most promising method in the sense that it simultaneously offers the highest number (376) of positive pairs and the lowest number (54) of negative pairs among its top-500 predicted PPIs, yielding a precision of 87.4% (see Fig. 5). The other two promising methods are MPS(T) and RNM, with precision 75.9% and 69.5%, respectively (see Supplementary Table 3 for the precision of other methods).

Note that in computational validation MPS(B&T) is ranked No.3 (in terms of the combined z-score in predicting human PPIs), while MPS(T) is ranked No.1, and RNM is ranked No.2. The ranking difference in computational and experimental validations is not a big surprise. This might be due to multiple reasons. First, in computational and experimental validations, methods were ranked based on different measures. In the computational validation, several performance measures were computed based on 10-fold cross-validation, and then different methods were ranked based on the combined z-score of the performance measures. In the experimental validation, only the top-500 predicted PPIs (leveraging the whole human interactome) of each method were validated and then the precision measures of different methods were ranked. Second, the different numbers of unsuccessfully tested PPIs in experimental validation might affect the performance ranking of different methods. For example, among the top-500 predicted PPIs, 70 142, or 106 pairs were not successfully tested for MPS(B&T), MPS(T), or RNM, respectively. Despite some changes in the ranking, the top-3 methods in computational validation, i.e., MPS(T), RNM and MPS(B&T) are still the top-3 methods in experimental validation. Only their relative rankings changed.

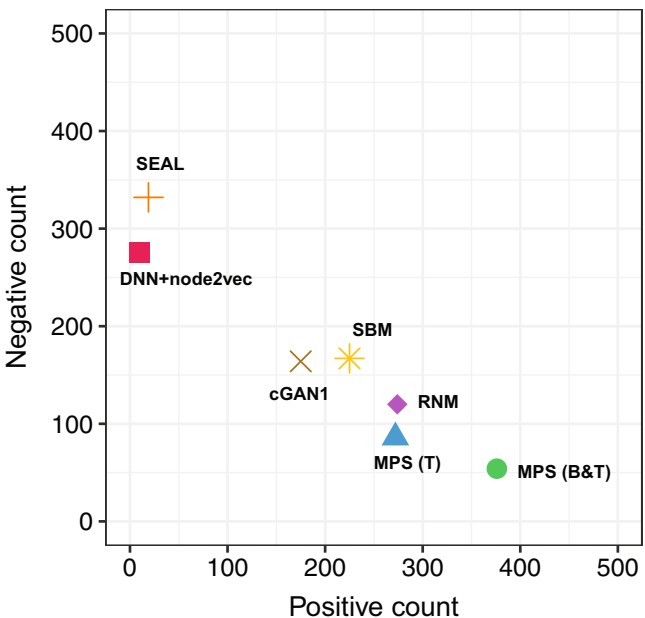

**Fig. 5 | Experimental evaluation of the top-seven human PPI prediction methods.** A protein pair is considered to be positive if it is positive in at least one of the three Y2H assays, and negative if it is negative in all the three assays. MPS(B&T) is the most promising method, which simultaneously offers the highest number (376) of positive pairs and the lowest number (54) of negative pairs among its top-500 predicted PPIs, yielding a precision of 87.4%. Note that the number of unsuccessfully tested protein pairs (e.g., due to a pipetting failure) is not included in the precision calculation and this figure. See Supplementary Table 3 for the positive count, negative count, unsuccessful test count, and the precision of other methods.

Interestingly, we found that those PPIs simultaneously predicted by multiple methods tend to be positive. For example, the 11 PPIs simultaneously predicted by RNM, MPS(T) and MPS(B&T) are all positive. (Note that those three methods all use the L3 principle in their own manner.) Yet, most of the positive PPIs were uniquely predicted by a particular method (see Supplementary Fig 10a for the Venn diagram). Another interesting aspect is that several tested methods showed experimental precision largely exceeding cross-validation results. While we suspect this might largely result from investigative biases reflected in available interactomes[43], exploring this aspect is beyond the scope of this work.

Note that the human interactome map HuRI contains self-loops, i.e., some proteins interact with themselves, representing the diagonal elements of the adjacency matrix of HuRI. We understand that the prediction of diagonal elements is orders-of-magnitude easier task than the prediction of off-diagonal elements in the adjacency matrix, due to the much larger density of self-interactions: In HuRI, the average degree of those self-interacting proteins is 35.05, while the mean degree of those non-self-interacting proteins is only 11.33. Among all prediction methods tested in this project, most of them tend to ignore self-loop prediction, but some of them (especially cGAN) do not. In fact, 495 of the top-500 PPIs predicted by cGAN are self-loops. We also found that those positive PPIs tend to connect high-degree proteins (see Supplementary Fig. 11).

## Combining predictions from the top three methods does not yield better precision

The predicted PPIs with higher ranking positions (i.e., in the top of the top-500 list) presumably should have higher probabilities of being positive in experimental validation than those predicted PPIs with a lower rank (i.e., in the bottom of the top-500 list). To test this assumption, for each of the top-three methods in human PPI prediction, we plotted the ranking position distribution of the

predicted PPIs that were validated to be positive in the Y2H experiments. As shown in Supplementary Fig. 12a, surprisingly, these positive PPIs do not tend to appear more often at the top of the list. Instead, they appear almost randomly in the top-500 PPIs predicted by each method. (It is unclear if this intriguing phenomenon will continue to hold if we test more pairs, e.g., top-1000 PPIs.) Consequently, combining the top-500 PPIs predicted by those top-ranking methods into a new top-500 list does not yield a better performance in experimental validation. To demonstrate quantitatively this point, we combined the top-$N_k$ PPIs predicted by MPS(B&T) and top-$[(500 - N_k)/2]$ PPIs from MPS(T) and RNM, respectively, with $N_k \in [0,500]$ defined as a tuning parameter. We ensured that those PPIs predicted by different methods appear only once in the combined list. We found that the number of positive PPIs monotonically decreases with $N_k$, indicating that combining the PPIs of greatest confidence predicted by different methods does not at all improve predictive performance (see Supplementary Fig. 12b).

## Structural and functional relationships of the validated and previously uncharacterized human PPIs

To explore the structural relationships of these predicted PPIs that tested positive in the Y2H assay, we visualized the network constituted by them (in total 1,177 PPIs involving 633 proteins), finding four distinct clusters (Fig. 6). These clusters were largely contributed by RNM, SBM, and MPS methods. We also found that the subnetworks contributed by RNM and SBM are close to each other. MPS(B&T) and MPS(T) contributes to a cluster together. In addition, MPS(B&T) contributed to a cluster itself. As shown in Supplementary Table 2, those methods leveraging the connectivity features, i.e., MPS, RNM, and SBM, tend to predict PPIs in dense neighborhoods (with higher edge density and shorter characteristic path length) of the interactome. By contrast, deep learning methods (e.g., SEAL, and DNN + node2vec) tend to predict PPIs that are more scattered in the interactome, and the induced subgraphs have lower edge density and longer characteristic path length. To quantify the distance between the proteins involved in the positive PPIs predicted by different methods, we computed their network-based separation[3] defined as $s_{\alpha\beta} = \langle d_{\alpha\beta} \rangle - (\langle d_{\alpha\alpha} \rangle + \langle d_{\beta\beta} \rangle)/2$, where $\alpha$ and $\beta$ represent the set of proteins involved in the positive PPIs predicted by two methods, respectively. $\langle d_{\alpha\beta} \rangle$ is the average shortest distance between proteins in $\alpha$ and $\beta$, $\langle d_{\alpha\alpha} \rangle$ (or $\langle d_{\beta\beta} \rangle$) is the average shortest distance between proteins within $\alpha$ (or $\beta$) in the original human interactome[4]. We found that almost all methods predicted PPIs in specific and separated areas of the interactome (Supplementary Fig. 10b), as each method is more likely to reflect different topological characteristics.

We also investigated the functional relationships of these positive PPIs, finding that they contribute to three functional modules, and each of those functional domains is associated with a distinctive, enriched GO term (Supplementary Fig. 13). This observation is also consistent with the previous finding that physical binding assembles proteins into large functional communities, thus providing insights into the global functional organization of the human cell[4]. Specifically, we found that the proteins involved in those top-500 predicted PPIs are involved in epidermis, e.g., cornification, keratin filament and keratinization and this is also valid for experimentally validated positive PPIs (see Supplementary Fig. 14a–c). For instance, previously uncharacterized PPIs associated with keratinization process from two dense clusters (see Supplementary Fig. 14d), which might be due to the fact that keratins and keratin association proteins are highly connected in HuRI. Of course, more function clusters will emerge if we focus on proteins involved in the top-5000 predicted PPIs. For example, when we analyzed the functions associated with the proteins in top-5000 PPIs predicted by MPS(B&T) and MPS(T), we found more function clusters, e.g., the one associated with extracellular exosome (see Supplementary Fig. 14).

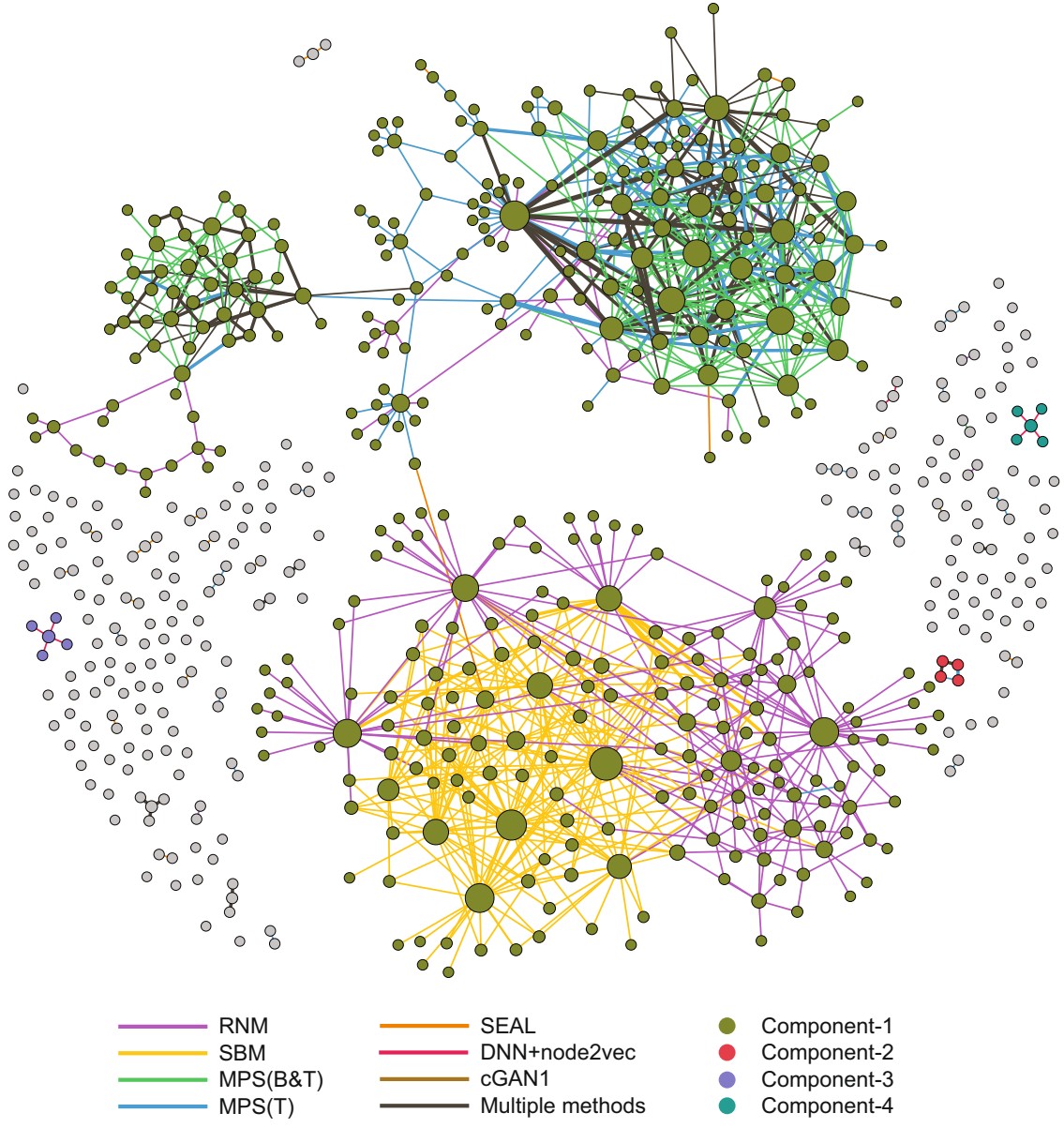

| | | |
|---|---|---|
| — RNM | — SEAL | ● Component-1 |
| — SBM | — DNN+node2vec | ● Component-2 |
| — MPS(B&T) | — cGAN1 | ● Component-3 |
| — MPS(T) | — Multiple methods | ● Component-4 |

**Fig. 6 | Structural relationships among previously uncharacterized human PPIs.** This network consists of all the 1177 previously uncharacterized human PPIs predicted by the top-seven methods and validated by Y2H assays. Those PPIs that were predicted by a single method were colored based on the method that predicted them. Those PPIs that were predicted (i.e., among the top-500 predicted PPIs) by multiple methods were colored in black, with edge width proportional to the number of methods predicting this PPI. Nodes (proteins) are colored based on the connected component to which they belong. Node size is proportional to its degree. Note that there are in total 174 isolated nodes, representing self-interacting proteins (which were mostly detected by cGAN1).

## Discussion

As knowledge of human PPIs can help us understand complex biological and disease mechanisms, developing computational algorithms to discover previously unrecognized PPIs and, thereby, to improve the comprehensiveness of the human interactome map is critical. To achieve this goal, we have evaluated 26 representative network-based PPI prediction methods across six different interactomes using 10-fold cross-validation. We then selected top-seven methods based on their performance in predicting PPIs in the human interactome. We applied each of the top-seven methods to the human interactome and predicted the top-500 unmapped human PPIs. We finally validated the union of the top-500 predicted human PPIs from the top-seven methods using the three orthogonal Y2H assays.

As a result of this systematic evaluation and validation effort, we identified the top-performing methods that prove useful for PPI prediction. Our analysis showed that the predictive power of traditional similarity-based methods is limited, although they are more easily portable without the need to rely on organism-specific annotations. Furthermore, generic link prediction methods based on deep machine learning methods, including embedding and graph neural networks approaches, performed consistently across different interactomes studied in this project with higher robustness, although their performances are not top-ranking. By contrast, link prediction methods MPS and RNM, which leverage specific connectivity properties of PPI networks (i.e., the L3 principle), displayed the most promising performance in the interactomes we considered. Similarity-based methods are already a promising way to guide protein–protein interaction assays, by prioritizing interactions with a high predicted score. More importantly, we found that different methods typically predicted positive PPIs that, rather than being scattered randomly in the interactome, are concentrated in specific areas (often associated with specific biological processes), and, furthermore, these areas overlap minimally among different methods.

**Table 1 | Computational Methods tested in the INMC PPI prediction project**

| ID Synopsis | U/S | Ref |
|---|---|---|
| **I. Similarity-based:** the existence probability of a link is measured as the prior knowledge-based similarity between two nodes ($i,j$). | | |
| 1. Common Neighbor (CN): similarity of a link is computed as the number of common neighbors between two nodes. | U | [61] |
| 2. Resource Allocation (RA): similarity of a link is computed as resource allocation of node pair. | U | [62] |
| 3. Preferential Attachment (PA): similarity of a link is computed as the degree product of node pair. | U | [63] |
| 4. Jaccard Index (JC): similarity of a link is computed as Jaccard index of node pair. | U | [64] |
| 5. Adamic Adar (AA): similarity of a link is defined as the Adamic-Adar index. | U | [65] |
| 6. Katz: similarity of a link is defined as the Katz index. | U | [66] |
| 7. Similarity (SIM): similarity score integrating L3 and Jaccard index. | U | [67] |
| 8. Ensemble: integrate several similarity scores. | U | |
| 9. Maximum similarity, Preferential attachment Score (MPS(T)): integrate two scores of topological features. | U | [68] |
| 10. Maximum similarity, Preferential attachment and sequence Score (MPS(B&T))*: integrate two scores of topological features and one score from the protein sequence. | U | [68] |
| 11. Root Noise Model (RNM): an ensemble method that integrates a diagonal noise model, a spectral noise model and L3 model. | U | [38] |
| 12. L3: paths of length three capture similarity to existing partners. | U | [38] |
| 13. CRA: similarity of a link is computed as CAR-based resource allocation. | U | [69] |
| **II. Probabilistic methods:** assume that real networks have some structure, i.e., community structure. The goal of these algorithms is to select model parameters that can maximize the likelihood of the observed structure. | | |
| 1. Stochastic Block Model (SBM): assume nodes are distributed into blocks and links between two nodes depends on the block they belong to. | U | [70] |
| 2. Repulsive Graph Signal Processing (RepGSP): learn PPIs via graph signal processing. RepGSP rewards links between "repulsive nodes" (i.e., nodes belonging to different communities). | U | [71–73] |
| **III. Factorization-based methods:** factorize the network adjacency matrix to reduce the high-dimensional nodes in the graph into a lower dimensional representation space by conserving the node neighborhood structures. | | |
| 1. Non-Negative Matrix Factorization (NNMF): dimension reduction using non-negative matrix factorization and the existence probability is defined as the cosine similarity of latent features. | U | [74] |
| 2. Geometric Laplacian Eigenmap Embedding (GLEE): dimension reduction using Geometric Laplacian Eigenmap, then defines existence probability as the cosine similarity of latent features. | U | [75] |
| 3. Spectral Clustering (SPC): dimension reduction using symmetric normalized Laplacian matrix, then the existence probability is defined as the cosine similarity of latent features. | U | [76] |
| **IV. Machine Learning:** methods based on machine learning techniques. | S | |
| 1. Conditional Generative Adversarial Network (cGAN): generative adversarial network performing image-to-image translation conditioned on either embedding (cGAN1) or raw information (cGAN2) of the network topology. | U | [57] |
| 2. Skip similarity Graph Neural Network (SkipGNN): receive neural messages from two-hop and immediate neighbors in the interaction network and non-linearly transforms the messages. | S | [17] |
| 3. Subgraphs, Embedding and Attributes for Link prediction (SEAL)*: learn general graph structure features from local enclosing subgraphs. | S | [16] |
| **V. Diffusion-based methods:** methods using techniques based on the analysis of the information diffusion over the network, e.g., random walks. This includes methods integrating techniques of other categories. | | |
| 1. Average Commute Time (ACT): similarity is defined as the average number of movements/steps required by a random walker to reach the destination node and come back to the starting node | U | [32] |
| 2. Random Walks with Restart (RWR): similarity is defined as the probability of a random walker node to reach the target node. | U | [77] |
| 3. Structural-Context Similarity (SimRank): measure the structural context similarity and shows object-to-object relationships. | U | [78] |
| 4. Deep Neural Network and Feature Representations for Nodes (DNN + node2vec): compute node and edge embeddings by the node2vec, then feeds the results into a deep neural network. | S | [58,79,80] |
| 5. Random Watcher-Walker (RW2)*: integrate network construction, network representation learning and classification. | U | [81] |

The U/S column is valued with U for unsupervised methods, otherwise S for supervised or semi-supervised methods. The * symbol indicates the level-2 methods, which make use of node or node-pair attributes.

This minimal overlap is due to the underlying assumptions of each method that highlight particular network patterns, suggesting that we may need to use different methods simultaneously to reflect the variable patterns in the interactome and offer complementary predictions. From a network perspective, it would also be interesting to analyze in more detail the association between biological processes and the structural patterns they express in the interactome, a research topic that we leave for future studies.

Encouragingly, the top-ranking methods were robust and seemed suitable for all the interactomes we studied in this project. However, we cannot comment on the applicability to link prediction in general as we validated these methods only on PPI networks rather than networks from different scientific domains. In our analysis, stacking models did not show higher performance than any individual method in PPI prediction, which could be attributed to the low overlap between PPIs predicted by different methods, as the overall search space is

enormous. For consistency, in this project we focused on reference interactomes generated from the Y2H system and the experimental validations were also conducted using the Y2H system, which is one of the most popular and powerful tools to study PPIs. Of course, there are other PPI-mapping techniques available, e.g., mass spectrometry[44]. We anticipate that the top-ranking methods presented here will still offer excellent performance in predicting PPIs for interactomes mapped by other techniques.

Despite their relatively high performance compared to other methods, deep learning-based methods (i.e., SEAL, SkipGNN, and DNN + node2vec) are not in the top three rank order in both computational and experimental validations. A reason for this failing could be the difference in the patterns of predicted PPIs, as remarked upon previously. For example, DNN + node2vec tends to predict PPIs involving proteins with lower degree than the top-three methods.

In this project, we focused on benchmarking 26 network-based methods covering different categories. Among the 26 methods, three of them (i.e., MPS(B&T), SEAL, and RW2) also leveraged protein sequence information. We were aware of those purely sequence-based PPI prediction methods (e.g., SVM[45–48], RF[49,50], FCTP[51], and DPPI[52]), as well as those methods leveraging additional biological information such as 3D protein structure and protein annotations[53,54]. We did not consider those methods in this benchmark study for two reasons. First, those methods need to define a feature space for each link, which will lead to significant time complexity and memory requirement for HuRI (which has ~35 million unmapped PPIs). Systematically benchmarking those methods is simply beyond the scope of the current project. Second, structural information has relatively little impact on constructing the interactomes, primarily because there is a great difference between the number of proteins with known sequences and those with an experimentally determined tertiary or quaternary structure[53]. In other words, this type of information is too incomplete to be efficiently exploited at the level of the entire interactome. The recent success of AlphaFold[55], a deep learning-based method to predict protein structure with atomic accuracy, is shedding light on resolving this limitation. In addition, we only considered six interactomes from four species, which certainly does not cover the variety and quality of all the available PPI datasets from different species.

Based on these findings, we recommend the following considerations for effective PPI prediction. First, the method needs to leverage the inherent properties of the interactome (e.g., the L3 principle) to improve the predictive performance. Second, the unmapped PPI space is over several hundred times larger than the currently mapped space, causing a limited overlap of the most probable PPIs predicted by different methods, which obviously reduces the efficacy of ensemble or stacking models. Finally, incorporating protein sequence and structure attributes into network-based methods appropriately could further improve the performance of PPI prediction, as soon as this type of information becomes available on a larger scale.

## Methods

### The INMC protein–protein interaction prediction project

This community effort was initiated by INMC aiming to provide a framework to assess the network-based computational methods in protein–protein interaction (PPI) prediction through standardized performance measures and common benchmarks. The INMC members were required to run their selected methods on six benchmark interactomes. For each interactome, members were required to submit two sets of results: (1) 10-fold cross-validation to compute the four performance measures; and (2) top-500 previously uncharacterized human PPIs predicted by their methods by leveraging the whole human interactome. In total, we tested 26 link prediction methods. The top-500 previously uncharacterized human PPIs provided by the top-7 high performance methods were further evaluated experimentally through yeast two-hybrid (Y2H) assays.

### PPI prediction methods

We compared in total 26 different methods that fall into five categories based on the adopted prediction strategy: similarity-based methods, probabilistic methods, factorization-based methods, machine learning methods, and diffusion-based methods (see Fig. 2). Based on the information used in the prediction, these methods can also be divided into two categories: level-1 (based on network structure only) and level-2 (based on both network structure and node attributes) (see Table 1). Based on the usage of training PPIs labels, they can be further divided into supervised and unsupervised methods. In the following, we will provide an overview of each category. All the methods are briefly described in Table 1. Additional details can be found in Supplementary Information (SI).

- Similarity-based methods: these link prediction methods use a similarity score function based on local properties of network nodes to measure the likelihood of links. Two nodes with higher similarity score are considered to have a link between them with higher probability. For example, two nodes with more common neighbors are considered to be more similar and tend to link to each other. Note that when applied to PPI prediction, the adopted similarity measures are mostly based on interconnection properties of the nodes rather than on specific node features. The main advantages of these methods are that they are agnostic to any annotation-dependent features, making them easily applicable across organisms; and that they make few assumptions about global network structure. We selected the classical similarity-based methods that were widely used in link prediction, i.e., CN, PA, RA, Katz and L3-based methods, i.e., MPS and RNM which showed superior performance in the prediction of PPIs.

- Probabilistic methods: The probabilistic and maximum likelihood algorithms assume that real networks have some structure, i.e., hierarchical or community structure. The goal of these algorithms is to select model parameters that can maximize the likelihood of the observed structure. As one of the most general network models, the stochastic block model (SBM) assumes that nodes are partitioned into groups with the probability that two nodes are connected depending solely on the groups to which they belong. We selected the classical probabilistic method SBM, and a new method RepGSP developed by ourselves.

- Factorization-based methods: These methods use matrix factorization techniques to find a mapping to embed the original dimensional nodes in the network into a lower dimension so that similar nodes in the original network tend to have similar representation features. The resulting embedded lower dimensional vectors (feature representations) can be used for many tasks, such as visualization, node classification, and link prediction. The link prediction task can be achieved by directly defining the likelihood of a link as the similarity of two nodes' embedded features or using other complex classifiers, e.g., linear regression or deep neural networks. We selected two classical factorization-based methods: NNFM, SPC, and a recently developed method GLEE.

- Machine learning: Machine learning (ML) is a growing field of pattern recognition algorithms that are trained on a given set of input data to make predictions based on the extracted patterns. Deep learning (DL) is a branch of machine learning composed of multi-layered neural network models. The recent success of deep neural networks is due to their ability to extract complex patterns in high-dimensional data by using non-linear functions. Graph neural networks (GNNs) are designed for learning over a graph. The graph convolution layers of the GNN are used to extract local substructure features for each node, and the graph aggregation layer aggregates node-level features into a graph-level feature vector[16]. These methods can learn parameters describing the general graph structural features and may include both node and connectivity features, showing promising performance in many network types[56]. We used two methods developed by ourselves (i.e., cGAN and DNN + node2vec), and two existing methods (SkipGNN and SEAL). The initial version of the cGAN (conditional generative adversarial network) method also utilizes features extracted via node2vec, which is a popular algorithmic framework for representational learning on graphs. The later refined version, cGAN2 relies solely on raw topological data[57]. Both SkipGNN and SEAL are GNN-based methods.

- Diffusion-based methods: These methods use techniques based on the analysis of the information gleaned from diffusion (typically from a random walker) over the network. We selected three widely used methods ACT, SimRank and RWR; and two additional methods developed by ourselves DNN + node2vec and RW2.

Details on the selected methods can be found in the SI. We note that most of the selected methods do not have hyperparameters. For those methods having hyperparameters (e.g., SEAL), we used their default or custom values without hyperparameter tuning. All surveyed methods use network (connectivity) information, while only a few (i.e., MPS(B&T), SEAL, RW2) incorporate information on protein sequence information.

### Performance metrics

We assessed the performance of each protein interaction prediction method using four metrics: Area Under the Receiver Operating Characteristic (ROC) curve (AUROC), Area Under the Precision-Recall Curve (AUPRC), Precision of the top-500 predicted PPIs (P@500), and Normalized Discounted Cumulative Gain (NDCG). Notice that we included AUROC since it is widely used in the link prediction literature[28–32] despite the fact that previous studies pointed out that AUROC is not a good performance metric for highly imbalanced data[26,27]. As the total number of PPIs for *C. elegans* and *S. cerevisiae* is less than 5000, the test PPIs in 10-fold cross-validation is less than 500, which means that the maximum P@500 is not 1. Given that the interactomes considered here are expected to be sparse, the number of true positive (i.e., existence) links will be dwarfed by the number of true negative (i.e., non-existent) links. In the literature, data imbalance is addressed by randomly selecting the same number of negative links to obtain a balanced validation list. Considering that we have also chosen three other metrics that can be used to quantify the classification methods and that are more robust to imbalanced data, we still reported the AUROC on the original imbalanced data.

Since the absolute values of AUROC and NDCG could be much larger than AUPRC and P@500, the values of AUPRC, P@500, and NDCG were each separately transformed into z-scores so that they have the same distribution, with mean value of 0 and standard deviation of 1. To compute a combined score that summarizes the performance of each method using the different evaluation metrics, we used the sum of the three z-scores, which is defined as:

$$\text{score} = z_{\text{AUPRC}} + z_{\text{P@500}} + z_{\text{NDCG}} \qquad (1)$$

### Evaluation strategy

To validate the methods, we applied two strategies: computational validation and experimental validation. Computational validation refers to a computational assessment of the 26 methods using the aforementioned performance metrics. Experimental validation evaluates the top-seven methods (criteria for the selection of which are described below) according to their rank in the computational validation by applying wet laboratory experiments on their predicted PPIs.

### Computational validation

For each of the 26 methods, we performed computational validation using the 10-fold cross-validation approach. We randomly split the observed link set $E$ into 10 subsets. For each iteration, one subset is selected as the probe set $E^P$ and links in this subset are removed from the network. Links in the remaining nine subsets constitute the training set $E^T$ and form the residual network. Note that some methods model both existing and non-existing links in training. For these methods, we added to the training set negative (non-existing) links generated by using balanced random sampling[58,59]. To compare different methods in

a systematic way, we computed the aforementioned performance metrics considering the test set as the union of the probe set $E^P$ and all the non-existing links not in the training set.

### Selection of the top-seven methods in human PPI prediction

Based on the results of the computational validation in the human PPI prediction, we selected the top-seven methods as follows. First, we ranked the 26 methods based on their combined z-scores, and focused on the top-10 methods: MPS(T), RNM, MPS(B&T), cGAN, SEAL, SBM, SkipGNN, CN, AA, and DNN + node2vec. L3[38] is already integrated in RNM, therefore it was not selected. Second, among the top-10 methods, we found that the top-6 methods (MPS(T), RNM, MPS(B&T), cGAN, SEAL, and SBM) have much higher combined z-scores than other methods, suggesting their superior performance over other methods. These six methods were selected for experimental validation. Finally, among the other four methods (SkipGNN, CN, AA, and DNN + node2vec), we found that their combined z-scores (as well as their AUPRC and NDCG) are quite close to each other. Considering the practical use case of PPI prediction methods, the P@500 metric might be the best performance measurement to estimate the applicability of a method in the real world, where only a portion of the predictions with the highest confidence scores will be experimentally evaluated. We selected DNN + node2vec as the seventh method to be validated experimentally, because its P@500 is much higher than that of the other three methods. Note that, we upgraded the RepGSP1 to RepGSP2, cGAN1 to cGAN2 after experimental validation, therefore RepGSP2 and cGAN2 were not selected though their performance is superior. For each of the selected seven methods (MPS(T), RNM, MPS(B&T), cGAN, SEAL, SBM, and DNN + node2vec), we validated its top-500 predicted PPIs based on the whole human interactome.

### Experimental validation

The union of the top-500 human PPIs predicted by the top-seven methods includes 3276 unique protein pairs. In addition, to benchmark the performance of the experiment, we tested a set of high-confidence binary PPIs curated from the published literature (PRS) and a set of pairs selected at random from the search space (RRS) (see Supplementary 9 for their recovery rate). We systematically tested those protein pairs by performing three complementary yeast two-hybrid (Y2H) assays. Briefly, 5 µl of glycerol stocks of Y8930:DB-ORF and Y8800:AD-ORF haploid strains were picked from the hORFeome collection into 200 µl of selective media (Synthetic Complete media without Leucine [SC-Leu] or Synthetic Complete media without Tryptophan [SC-Trp], respectively) and arrayed to generate the pairs to be tested. After overnight growth, 5 µl of Y8930:DB-ORF and 5 µl of Y8800:AD-ORF culture were transferred into YEPD (Yeast Extract Peptone Dextrose). After incubating overnight at 30 °C to allow mating to occur, 10 µl of yeast culture was transferred into 120 µl SC-Leu-Trp media to allow for selection of diploid yeast cells. The next day, diploid yeast cultures were spotted on Synthetic Complete media without leucine, tryptophan and histidine with 1 mM 3-Amino-1,2,4-triazole (SC-Leu-Trp-His+1 mM 3AT) to test for interactions and SC-Leu-His +1 mM 3AT supplemented with either 1 mg/L cycloheximide (CHX) for assay version 1 or 10 mg/L for assay versions 2 and 3 to test for auto-activation. After 72 h incubation at 30 °C, diploid cells that grew on SC-Leu-Trp-His+3AT media but not on SC-Leu-His+3AT + CHX media were scored positive. If a pair had a similar or higher frequency of yeast colonies on the CHX plate compared to the test plate, then it was scored as a spontaneous auto-activator (AA). Cases showing contamination or where no diploid yeast were spotted (for example due to a pipetting failure), were reported as "not-tested". Experiments were only considered complete where both controls performed as expected. Plasmid details are described in ref. [4] and http://www.interactome-atlas.org.

**Reporting summary**

Further information on research design is available in the Nature Portfolio Reporting Summary linked to this article.

## Data availability

All data are available at https://github.com/spxuw/PPI-Prediction-Project. Source data are provided with this paper.

## Code availability

All data, Python and Matlab scripts developed by authors are available at https://github.com/spxuw/PPI-Prediction-Project or under Zenodo at https://doi.org/10.5281/zenodo.7681817[60], where the links to all other publicly available methods are also provided.

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

## Acknowledgements

L.M., A.F., and L.B. were partially supported by the ERC Advanced Grant 788893 AMDROMA "Algorithmic and Mechanism Design Research in Online Markets", the EC H2020RIA project "SoBigData++" (871042), and the MIUR PRIN project ALGADIMAR "Algorithms, Games, and Digital Markets". F.L. was supported by a Wallonia-Brussels International (WBI)-World Excellence Fellowship, a Fonds de la Recherche Scientifique (FRS-FNRS)-Télévie Grant (FC31747, Crédit no. 7459421F), a Herman-van Beneden Prize and a Léon Frédéricq Foundation-Josée & Jean Schmets Prize. M.V. is a Chercheur Qualifié Honoraire from the Fonds de la Recherche Scientifique (FRS-FNRS, Wallonia-Brussels Federation, Belgium). M.V acknowledges support from the National Institute of Health (R01 GM130885). P.F. and B.Á. were supported by the National Research, Development and Innovation Office of Hungary (National Heart Program NVKP 16-1-2016-0017) and the Thematic Excellence Programme (2020-4.1.1.-TKP2020) of the Ministry for Innovation and Technology in Hungary, within the framework of the Therapeutic Development and Bioimaging thematic programmes of the Semmelweis University. Project no. RRF-2.3.1-21-2022-00003 has been implemented with the support provided by the European Union. JL acknowledges support from the National Institutes of Health (R01 HL155107, R01 HL155096, U01 HG007690, and U54 HL119145); and from the American Heart

Association (D700382 and CV-19). A-LB is supported by the Veteran's Affairs Medical Center of Boston Contract #36C24122N0769, the NIH grant #1P01HL132825 And the European Union's Horizon 2020 research and innovation programme under grant agreement No 810115 – DYNASNET. Y.-Y.L. acknowledges grants from National Institutes of Health (R01AI141529, R01HD093761, RF1AG067744, UH3OD023268, U19AI095219, and U01HL089856).

## Author contributions

Y.-Y.L. and P.V. conceived and designed the project. L.M., A.F. and L.B. developed and tested the MPS(T) and MPS(B&T) methods. T.W. and I.K. developed and tested the RNM method. O.M.B., B.B., M.P., B.Á., and P.F. developed and tested the cGAN method. L.V. and J.M. developed and tested the DNN+node2vec method. S.C., M.P., G.S., and F.C. developed and tested the RepGSP method. L.M. tested the RW method. X.-W.W. tested the other 19 methods. K.S., T.H., F.L., L.W., J.-C.T., and M.A.C. conducted the experimental validations. X.-W.W., L.M., P.V. and Y.-Y.L. analyzed the results. X.-W.W. and Y.-Y.L. wrote the manuscript with assistance from L.M., P.V., and K.S., L.B., I.K., B.Á., L.V., S.C., M.A.C., A.-LB., E.K.S., and J.L. edited the manuscript. All authors approved the final manuscript.

## Competing interests

PF is the founder and CEO of Pharmahungary Group, a group of R&D companies. EKS has received institutional grant support from Bayer and GlaxoSimthKline. A-LB is co-scientific founder of and is supported by Scipher Medicine, Inc., which applies network medicine strategies to biomarker development and personalized drug selection, and is the founder of Naring Inc., which applies data science to health and nutrition. The remaining authors declare no competing interests.

## Additional information

[1]Channing Division of Network Medicine, Department of Medicine, Brigham and Women's Hospital and Harvard Medical School, Boston, MA 02115, USA. [2]Translational and Precision Medicine Department Sapienza University of Rome, Rome, Italy. [3]Center for Cancer Systems Biology (CCSB), Dana-Farber Cancer Institute, Boston, MA 02215, USA. [4]Department of Genetics, Blavatnik Institute, Harvard Medical School, Boston, MA 02115, USA. [5]Department of Cancer Biology, Dana-Farber Cancer Institute, Boston, MA 02215, USA. [6]Department of Computer, Control, and Management Engineering "Antonio Rubert", Sapienza University of Rome, Rome, Italy. [7]CENTAI Institute, Turin, Italy. [8]Department of Physics and Astronomy, Northwestern University, Evanston, IL 60208, USA. [9]Northwestern Institute on Complex Systems, Northwestern University, Evanston, IL 60208, USA. [10]Cardiometabolic and MTA-SE System Pharmacology Research Group, Department of Pharmacology and Pharmacotherapy, Semmelweis University, Budapest, Hungary. [11]Pharmahungary Group, 6722 Szeged, Hungary. [12]CeMM Research Center for Molecular Medicine of the Austrian Academy of Sciences, Vienna, Austria. [13]Department of Structural and Computational Biology, Max Perutz Labs, University of Vienna, Vienna, Austria. [14]Faculty of Mathematics, University of Vienna, Vienna, Austria. [15]Department of Information Engineering, Electronics, and Telecommunications (DIET), University of Rome "Sapienza", Rome, Italy. [16]Laboratory of Molecular and Cellular Epigenetic, GIGA Institute, University of Liège, Liège, Belgium. [17]Laboratory of Viral Interactomes, GIGA Institute, University of Liège, Liège, Belgium. [18]TERRA Teaching and Research Centre, University of Liège, Gembloux, Belgium. [19]Department of Medicine, Brigham and Women's Hospital and Harvard Medical School, Boston, MA 02115, USA. [20]Department of General Internal Medicine and Primary Care, Brigham and Women's Hospital, Boston, MA 02115, USA. [21]Network Science Institute and Department of Physics, Northeastern University, Boston, MA 02115, USA. [22]Department of Network and Data Science, Central European University, Budapest H-1051, Hungary. [23]Center for Artificial Intelligence and Modeling, The Carl R. Woese Institute for Genomic Biology, University of Illinois at Urbana-Champaign, Champaign, IL 61801, USA. ✉e-mail: velardi@di.uniroma1.it; yyl@channing.harvard.edu

