## [Peer Review File · Nature Communications]

Reviewers' Comments:

Reviewer #1:

Remarks to the Author:

The manuscript by Xu-Wen Wang and his colleagues describes a community effort to benchmark 24 representative network-based PPI prediction methods across multiple model organisms, *Homo sapiens* and a synthetic network. The work focuses on predicting co-complex binary interaction data. The initial networks used in the benchmark were derived from an experimental (Y2H) data with varied proteome coverage and network structure. The methods were benchmarked using 10-fold cross-validation. The authors singled out 7 best performing methods based various statistics including AUROC, AUPRC, True Positive Rate in top 500 interaction and few others. Subsequently the authors created a combined set of top-500 prediction from each of the 7 methods resulting in 3,276 PPIs from 633 human proteins. This combined set was experimentally validated using Y2H method yielding a relatively high level (1,177 or around 35%) of positively validated pairs.

My first question relates to the difference seen between AUROC and AUPRC statistics. I cannot think of a curve with almost perfect (~ 0.9) AUROC and almost non-existent AUPRC (~ 0.01) for the same set of predictions and the same benchmark. If the recall is bad, which is common with large positive sets (such as full PPI network) the AUROC should be quite close to 0.5 even if there is a meaningful performance on the top end of the predictions. This is due to the fact that we do not expect high-recall for such a large search-space. However this is not what we see as high AUROC implies high recall at the top end. By contrast low AUPRC must indicate low precision, but then it should be visible in high False Positive Rate lowering the overall AUROC. Something does not add up here. Overall it would be much easier to evaluate the methods if the authors would provide all the data available predictions for all the methods rather than the combined set and the simple statistical summary.

According to the authors the negative set consists of "all the non-existing links not in the training set". This would amount in the HuRI dataset to around $\sim 32,000,000$ (fully connected graph of 8,000 proteins) negative pairs. Given AUROC of 0.9 on such an unbalanced benchmark (50,000+ positive pairs) would suggest an almost perfectly solved human PPI network which leads me to believe it is not how any of this was calculated. The authors should clearly provide information how to derive that data and, again, provide all the underlying predictions for all the methods, together with the benchmark status, so that anyone can draw the ROC and PR curve for each of the methods.

HuRI dataset which authors use for the positive set was made with 3 different version of Y2H in triplicates. The authors used the same method, just without the replicates. Given the inherent bias of Y2H towards certain kind of interactions, I'm not 100% this is the best spent effort for what would amount to another (arguably targeted) replicate. There should be at least one additional validation using any other available PPI data from other resources and methods such as IntACT or BioGrid.

Considering so much effort was spend on experimentally validating the results it is surprising the authors do not elaborate about the predicted interactions. The only dataset that was provided is the combined top 500 prediction form the 7 best performing methods (in total 3,276 pairs) and their experimental validation status. What is striking is that around 750 of pairs (or around 25%) relates to Keratin or Keratin-associated protein or Late Cornfield Envelope proteins, all of them associated with epidermis and potentially form a functional cluster. More than half of these pairs are solely between these 3 types of proteins, most of which are positive (this amounts to 25% of all positive pairs). In my opinion this could indicate a bias in the prediction(s) method (as they share interaction partners and thus the network signal) and possibly in Y2H assay. I would like the authors to elaborate what would be the reason for finding so many epidermis associated proteins.

As one could reasonably argue this could be correct, therefore I've looked at what is there apart from the these 3 types of proteins. Alphabetically first protein, with more than one interaction in the 3276 protein pairs set is ABCE1. According to the predictions It interacts with CSNK1E, CYB5D2, OTOS, PLP2. As far as I can find there is nothing that links these proteins. Not physically or functionally.

Second prediction is for ABI1 which interacts with ARL13B, C3orf33, CRY2, FBXL3, KCNS2, NFATC2IP, PLP2, PSMD7, TEK4, TMEM50A. From this set FBXL3 and CRY2 form a complex with each other, so there is something to it, but I do not see ABI1 forming a complex with any of the proteins. As for the validated subset: first protein (alphabetically) that has several experimentally validated interaction is APOL2, which interacts with validated targets ELOVL4, ERGIC3, FAM210B, SLC35C2, TIMMDC1. There is no process that I can see that links these protein. Even disregarding if they form a complex with APOL2 or not, one would expect some functional relatedness between the interaction partners of APOL2.

Given that I am somewhat suspicious about both the predictions and the validation regime of the manuscript. I'm sympathetic to the premise of the manuscript, which is the need for systematic evaluation of the network prediction methods. The author admit as much as "predictability of interactomes is in general weak", however rather than offering insight why that would be case or exploring pitfalls of using these method we see an effort in trying to convince the audience that these method have merit. I don't see it, neither on a conceptual level nor in the presented data. I could be wrong of course.

I suggest that the manuscript should be substantially revised. Primarily a) All the predictions should be made available, b) additional benchmarking on orthogonal datasets should be done and c) discussion and insight in the predictions and the methods's potential biases, if it's not random, and seems to be barely functional (sans the keratin), what signal do they actually capture?

Reviewer #2:

Remarks to the Author:

In this manuscript, the authors evaluated 24 network-based PPI prediction methods on four interactomes from *A. thaliana*, *C. elegans*, *S. cerevisiae*, and *H. sapiens*, respectively, and one synthetic interactome using 10-fold cross validation. Based on cross validation results from the human interactome, the top-seven methods were selected, and the top-500 unmapped human PPIs predicted by each of these methods were experimentally validated using three orthogonal Y2H assays. In both cross validation and experimental validation, three methods, Maximum similarity and Preference attachment Score MPS(T), Maximum similarity, Preference attachment and sequence Score (MPS(B&T)), and Root Noise Model (RNM) showed the best performance.

The analysis is reasonable, and the results confirm the reports from Reference #69 and Reference #36. Although this study includes more methods in the benchmarking compared with the published studies, methodological and conceptual innovation is limited. One potentially interesting observation is that some deep learning based method performed ok in cross validation but failed badly in experimental validation, but there is very limited investigation on this observation. I also have some other comments:

1. P@500 is very arbitrary, AP@K would be a more robust metric.
2. Figure S1, a-c, pearson's correlation is very misleading, spearman should be used.
3. Figure 3a, it would be nice to leave a gap between different method groups. Also separate traditional similarity based methods from advanced similarity based methods as this is the major conclusion from the paper.
4. The main text in Page 5 mentioned Fig.3a-e, but there are only two panels in Figure 3
5. Figure S2, the *H. Sapiens* network showed the lowest predictability but it is associated with the best performance in Figure 3, this needs to be explained.
6. It would be nice to have a figure simultaneously visualizing average rankings of the 24 methods on the five networks and ranking variability
7. The section "there are four consistently high-performing methods", why cGAN is excluded? It looks good in Figure 3b.
8. It seems that degree plays a major role in many algorithms. To investigate degree-driven

prediction, it would be interesting to apply the prediction methods to networks generated by degree-preserving rewiring and check for their prediction performance.

9. Figure S8, it is strange to perform functional analysis of the predicted PPIs without putting them in the context of the known PPIs used for the prediction.

10. In experimental validation, is there a trend that predicted PPIs involving high-degree proteins are more likely to be validated?

Reviewer #3:

Remarks to the Author:

Summary.

Protein interactomes, that are often modeled as protein-protein interaction (PPI) networks, provide important insights into the functioning of the cell, which is key for understanding the molecular bases of biological processes and diseases. However, due to cost of experimentally capturing PPIs, interactomes remain largely unmapped. To overcome this limitation, many PPI prediction methods have been proposed in the literature.

In this paper, the authors present a systematic evaluation of the performances of network-based PPI prediction approaches, and also incorporate large-scale confirmatory experimentation (wet-lab).

- On real and synthetic interactomes, they evaluate the prediction performances of 24 state of the art methods. A key result is that the predictability of interactome is low, which may be due to the incompleteness of the PPI data. The authors also highlight four methods that consistently perform well, namely RNM, MPS(T), MPS(B&T), and SEAL.

- Then, the authors focus on the seven best performing approaches and further validate their top-500 PPI predictions on the human interactome using wet-lab experiments (using Y2H assay). From the corresponding 3,276 unique predictions, they validated 1,177 PPI between 633 proteins.

Assessment.

Overall, the paper is well written, and the proposed methodology to compare different PPI prediction methods is sound (it properly uses cross-fold validation and 4 different performance metrics), and the large scale wet-lab validation of the top predicted PPI is impressive. However, there are some issues.

First, unlike claimed, this is not the first time that a large scale experimental validation is conducted to confirm the computational predictions of network based PPI prediction methods (e.g., see [1]). However, the scale of the wet-lab validation is impressive.

Second, and more importantly, the computational comparative analysis is plagued by a poor choice of PPI data on which it is performed. As a consequence, the presented results and conclusions may not 'generalize' to the current wealth of PPI data available (more species, larger sizes, and larger variety of capturing technologies).

In particular, to avoid biases, the authors decided to cherry pick high quality PPI datasets. This leads to the following issues:

1- The computational comparison is based on only five (four real and one synthetic) interactomes, which is underwhelming and surprising given that the authors compared a large number of methods (24) and did a large scale wet-lab validation of the top-ranked PPI predictions. With only four real interactomes, the dataset is not representative of the large variety of available PPI data; e.g., there are far more interactome data available in databases such as BioGRID [2] or STRING [3]. For instance, in [1], the performance of L3-based PPI prediction method is evaluated on 16 interactomes

from 7 species.

2- Also, the selected interactomes are very small / very incomplete, and may not represent the completeness of available PPI data (e.g., the human PPI data used in the study has 52K interactions between 8K proteins, while current human experimental PPI network from BioGRID database has 521K interactions between 19K proteins).

3- Despite the claimed objective of reducing biases in the data, three out of the four real interactomes (C. elegans, yeast, and human) are captured using the same technology (Y2H), so a strong technological bias remains.

To overcome these limitations, the authors should include more interactomes, captured with a larger variety of bio-technologies, and with varying amount of experimental validation.

As a minor comment, I don't see the benefit of using only one synthetic interactome. Often, random model network generators are used to generate 'controlled' datasets to study the effect of a given parameter (e.g., the number of nodes, the number of interaction edges, the amount of noise, or of the underlying network models), e.g., in the context of network comparison [4]. But with only one network, none of this is possible. For instance, the authors could have used synthetic networks generated with varying edge densities to further support their claim that more complete interactomes (i.e., with larger edge densities) are more predictable.

[1] Kovács et al. (2019). Network-based prediction of protein interactions. Nature communications, 10(1), 1-8.

[2] BioGRID database statistic: <https://wiki.thebiogrid.org/doku.php/statistics>

[3] STRING database statistics: https://string-db.org/cgi/about?sessionId=b9wz9kS1gbKk&footer_active_subpage=statistics

[4] Yaveroglu et al. (2015), Proper evaluation of alignment-free network comparison methods, Bioinformatics, 31(16):2697-2704.

Responses to Reviewer #1

Point 1.0: The manuscript by Xu-Wen Wang and his colleagues describes a community effort to benchmark 24 representative network-based PPI prediction methods across multiple model organisms, Homo sapiens and a synthetic network. The work focuses on predicting co-complex binary interaction data. The initial networks used in the benchmark were derived from an experimental (Y2H) data with varied proteome coverage and network structure. The methods were benchmarked using 10-fold cross-validation. The authors singled out 7 best performing methods based various statistics including AUROC, AUPRC, True Positive Rate in top 500 interaction and few others. Subsequently the authors created a combined set of top-500 prediction from each of the 7 methods resulting in 3.276 PPIs from 633 human proteins. This combined set was experimentally validated using Y2H method yielding a relatively high level (1,177 or around 35%) of positively validated pairs.

Response: We thank Reviewer #1 for reviewing our manuscript. Next, we address each of the reviewer's comments in order.

Point 1.1: My first question relates to the difference seen between AUROC and AUPRC statistics. I cannot think of a curve with almost perfect (~0.9) AUROC and almost non-existent AUPRC (~0.01) for the same set of predictions and the same benchmark. If the recall is bad, which is common with large positive sets (such as full PPI network) the AUROC should be quite close to 0.5 even if there is a meaningful performance on the top end of the predictions. This is due to the fact that we do not expect high-recall for such a large search-space. However this is not what we see as high AUROC implies high recall at the top end. By contrast low AUPRC must indicate low precision, but then it should be visible in high False Positive Rate lowering the overall AUROC. Something does not add up here.

Response: We thank Reviewer #1 for this critical comment. The big difference between the high AUROC value and the low AUPRC value is due to the fact that the PPI network is highly imbalanced, i.e., the number of negative links (35 million for HuRI) is significantly higher than the number of positive ones (52,548 for HuRI). In the context of link prediction, AUROC corresponds to the probability that prediction scores from a randomly selected pair of positive and negative links are correctly ordered, while AUPRC reflects the ability of using the prediction scores to identify the positive links. For highly imbalanced dataset, AUROC will still be good enough even if the link prediction method misclassified most (or all) of the minority group (i.e., the group of positive links in the PPI network); while AUPRC will generally be much lower than AUROC because identifying few positive links from a huge search space is intrinsically challenging.

To clearly demonstrate this point, in **Figure R1**, we showed two examples, one represents a highly imbalanced dataset (with 50 positive instances and 100,000 negative instances), while the other represents a balanced dataset (with 50 positive instances and 50 negative instances). In both cases, scores of positive instances were generated from a normal distribution with mean 0.15 and standard deviation 0.1; while scores of negative instances were generated from a normal distribution with mean 0 and standard deviation 0.1. As shown in Fig.R1, for the first example, we have AUROC= 0.86 and AUPRC=0.008; while for the second example, we have AUROC=0.89 and AUPRC=0.91. Raw code and data used in this demonstration have been posted on our GitHub webpage: <https://github.com/spxuw/PPI-Prediction-Project>.

Point 1.2: Overall it would be much easier to evaluate the methods if the authors would provide all the data available predictions for all the methods rather than the combined set and the simple statistical summary.

Response: We thank Reviewer #1 for this very critical comment. The reason why we did not provide the intermediate results (i.e., prediction scores) is due to the large storage space requirement. For example, the BioGRID interactome has 19,665 nodes, so the three-column prediction list file has $193,346,280 \times 3$ numbers, where the first and second columns represent the source and target nodes, respectively, and the third column is the prediction score. For each interactome, we performed 10-fold validation using 26 prediction methods. This will lead to a huge storage challenge (more than 3.5T) just for this interactome.

To ensure all readers can easily reproduce our results, we have made the code of all the methods publicly available through our GitHub webpage: <https://github.com/spxuw/PPI-Prediction-Project>. To further ensure reproducibility (but without causing a serious storage issue), we also posted the link of three-column prediction list file from applying one of the best methods, RNM, to a human interactome (HuRI), with one particular training/test split, on our GitHub webpage.

Point 1.3: According to the authors the negative set consists of “all the non-existing links not in the training set”. This would amount in the HuRI dataset to around ~32.000.000 (fully connected graph of 8.000 proteins) negative pairs. Given AUROC of 0.9 on such an unbalanced benchmark (50.000+ positive pairs) would suggest an almost perfectly solved human PPI network which leads me to believe it is not how any of this was calculated. The authors should clearly provide information how to derive that data and, again, provide all the underlying predictions for all the methods, together with the benchmark status, so that anyone can draw the ROC and PR curve for each of the methods.

Response: We thank Reviewer #1 for this comment. As we mentioned in the main text “This very high AUROC might lead us to mistakenly conclude that SEAL is an almost perfect prediction method since the maximum value of AUROC is 1”. Again, this is due to the imbalance issue of link prediction for PPI networks, which has been explained in our response to Point 1.1 and demonstrated in **Figure R1**.

Point 1.4: HuRI dataset which authors use for the positive set was made with 3 different version of Y2H in triplicates. The authors used the same method, just without the replicates. Given the inherent bias of Y2H towards certain kind of interactions, I’m not 100% this is the best spent effort for what would amount to another (arguably targeted) replicate. There should be at least one additional validation using any other available PPI data from other resources and methods such as IntACT or BioGrid.

Response: We thank Reviewer #1 for this very constructive comment. In the revised manuscript, we have evaluated those methods on two additional human PPI databases: BioGRID and STRING (see **Figure R2**). We found that our previous conclusions (i.e., predictability of interactomes is weak, most methods vary considerably across different interactomes, traditional similarity-based methods do not perform well, etc.) still hold.

Point 1.5: Considering so much effort was spent on experimentally validating the results it is surprising the authors do not elaborate about the predicted interactions. The only dataset that

was provided is the combined top 500 prediction from the 7 best performing methods (in total 3,276 pairs) and their experimental validation status. What is striking is that around 750 of pairs (or around 25%) relates to Keratin or Keratin-associated protein or Late Cornfield Envelope proteins, all of them associated with epidermis and potentially form a functional cluster. More than half of these pairs are solely between these 3 types of proteins, most of which are positive (this amounts to 25% of all positive pairs). In my opinion this could indicate a bias in the prediction(s) method (as they share interaction partners and thus the network signal) and possibly in Y2H assay. I would like the authors to elaborate what would be the reason for finding so many epidermis associated proteins.

Response: We thank Reviewer #1 for this very insightful comment. To examine the predicted interactions, we first analyzed the functions associated with the proteins involved in the top-500 PPIs predicted by the two top methods MPS(T) and MPS(B&T). The results (in terms of the number of proteins associated with each GO) are shown in **Figure R3a,b**. We did find three keratin-related GOs, i.e., keratinization, keratin filament and cornification.

We suspect that those specific function clusters might be simply due to the fact that we focused on proteins involved in the top-500 predicted PPIs and those highly confident PPIs form a dense cluster (see main text, Figure 6). When we analyzed the functions associated with the proteins in top-5000 PPIs predicted by MPS(B&T) and MPS(T), we did find some new function clusters, e.g., the one associated with extracellular exosome (colored in blue, **Figure R3c,d**).

In the revised manuscript, we have included Fig.R3 as Fig.S13, and added the following sentences to the main text (see page 10, lines 393-400):

“Specifically, we found that the proteins involved in those top-500 predicted PPIs are involved in epidermis, e.g., cornification, keratin filament and keratinization and this is also valid for experimentally validated positive PPIs (see Fig.S13 a-c). For instance, new PPIs associated with keratinization process from two dense clusters (see Fig.S13d). Of course, more function clusters will emerge if we focus on proteins involved in the top-5000 predicted PPIs. For example, when we analyzed the functions associated with the proteins in top-5000 PPIs predicted by MPS(B&T) and MPS(T), we found new function clusters, e.g., the one associated with extracellular exosome (see Fig.S13).”

Point 1.6: As one could reasonably argue this could be correct, therefore I've looked at what is there apart from the these 3 types of proteins. Alphabetically first protein, with more than one interaction in the 3276 protein pairs set is ABCE1. According to the predictions It interacts with CSNK1E, CYB5D2, OTOS, PLP2. As far as I can find there is nothing that links these proteins. Not physically or functionally.

Response: We thank Reviewer #1 for this comment. Those 3,276 pairs were predicted by 7 different methods and each method highlights different structure in the PPIs. Moreover, some of the predicted pairs have been shown to be negative using the Y2H assay. For example, protein ABCE1 (ENSG00000164163) was predicted to interact with CSNK1E (ENSG00000213923), CYB5D2 (ENSG00000167740), OTOS (ENSG00000178602), and PLP2 (ENSG00000102007). However, those four pairs have been shown to be negative through our experimental validation.

Point 1.7: Second prediction is for ABI1 which interacts with ARL13B, C3orf33, CRY2, FBXL3, KCNS2, NFATC2IP, PLP2, PSMD7, TEKT4, TMEM50A. From this set FBXL3 and CRY2 form a

complex with each other, so there is something to it, but I do not see API1 forming a complex with any of the proteins.

Response: We thank Reviewer #1 for this comment. The proteins that were predicted to interact with ABI1 (ENSG00000136754) are: ARL13B (ENSG00000169379), C3orf33 (ENSG00000174928), CRY2 (ENSG00000121671), FBXL3 (ENSG00000005812), KCNS2 (ENSG00000156486), NFATC2IP (ENSG00000176953), PLP2 (ENSG00000102007), PSMD7 (ENSG00000103035), TEKT4 (ENSG00000163060) and TMEM50A (ENSG00000183726). However, those ten pairs have also been shown to negative through our experimental validation.

Point 1.8: As for the validated subset: first protein (alphabetically) that has several experimentally validated interaction is APOL2, which interacts with validated targets ELOVL4, ERGIC3, FAM210B, SLC35C2, TIMMDC1. There is no process that I can see that links these protein. Even disregarding if they form a complex with APOL2 or not, one would expect some functional relatedness between the interaction partners of APOL2.

Response: We thank Reviewer #1 for this critical comment. Indeed, the validated subset of proteins that interact with protein APOL2 (ENSG00000128335) includes: ELOVL4 (ENSG00000118402), ERGIC3 (ENSG00000125991), FAM210B (ENSG00000124098), SLC35C2 (ENSG00000080189), and TIMMDC1 (ENSG00000113845). We noticed that those six proteins are associated with the GO term: organelle membrane. In addition, some genes (e.g., APOL2) are poorly studied, so relevant functional annotation could be missing, and the existing PPI network patterns seem sufficient to suggest new interactions.

Point 1.9: Given that I am somewhat suspicious about both the predictions and the validation regime of the manuscript. I'm sympathetic to the premise of the manuscript, which is the need for systematic evaluation of the network prediction methods. The author admit as much as "predictability of interactomes is in general weak", however rather than offering insight why that would be case or exploring pitfalls of using these method we see an effort in trying to convince the audience that these method have merit. I don't see it, neither on a conceptual level nor in the presented data. I could be wrong of course.

Response: We thank Reviewer #1 for this critical comment. The weak predictability of the interactomes is due to the possible PPI space is extremely large. For instance, HuRI has more than 8,000 proteins, hence the total possible number of links is ~3,200,000. Identifying true PPIs from such a huge space is a challenging task. But this does not mean that we cannot make any progress using computational methods. Indeed, our results demonstrated that advanced similarity-based methods show superior performance over other general link prediction methods. Through experimental validation, we confirmed that the top-ranking methods show promising performance externally, often better than computational validation would suggest. For example, from the top-500 PPIs predicted by an advanced similarity-based method [MPS(B&T)], 430 were successfully tested by Y2H with 376 testing positive, yielding a precision of 87.4%. These results suggest that advanced similarity-based methods are powerful tools for the prediction of human PPIs.

In the revised main text, we have added the following sentence regarding this point (see page 11. lines 419-421):

"Similarity-based methods are already a promising way to guide protein-protein interaction assays, by prioritizing interactions with a high predicted score."

Point 1.10: I suggest that the manuscript should be substantially revised. Primarily a) All the predictions should be made available, b) additional benchmarking on orthogonal datasets should be done and c) discussion and insight in the predictions and the methods's potential biases, if it's not random, and seems to be barely functional (sans the keratin), what signal do they actually capture?

Response: We thank Reviewer #1 again for reviewing our manuscript and her/his very insightful comments. We have explained the very high values of AUROC in our response to Point 1.2; performed additional benchmarking on two orthogonal PPI datasets: BioGRID and STRING; and explained the prediction bias in our response to Point 1.5. We hope our responses have addressed those comments in a satisfactory manner.

Responses to Reviewer #2

Point 2.0: In this manuscript, the authors evaluated 24 network-based PPI prediction methods on four interactomes from *A. thaliana*, *C. elegans*, *S. cerevisiae*, and *H. sapiens*, respectively, and one synthetic interactome using 10-fold cross validation. Based on cross validation results from the human interactome, the top-seven methods were selected, and the top-500 unmapped human PPIs predicted by each of these methods were experimentally validated using three orthogonal Y2H assays. In both cross validation and experimental validation, three methods, Maximum similarity and Preference attachment Score MPS(T), Maximum similarity, Preference attachment and sequence Score (MPS(B&T)), and Root Noise Model (RNM) showed the best performance.

The analysis is reasonable, and the results confirm the reports from Reference #69 and Reference #36. Although this study includes more methods in the benchmarking compared with the published studies, methodological and conceptual innovation is limited. One potentially interesting observation is that some deep learning based method performed ok in cross validation but failed badly in experimental validation, but there is very limited investigation on this observation. I also have some other comments:

Response: We thank Reviewer #2 for reviewing our manuscript. Next, we address each of the reviewer's comments in order.

Point 2.1: 1. P@500 is very arbitrary, AP@K would be a more robust metric.

Response: We thank Review #2 for this constructive comment. The reason of choosing P@500 as a performance metric is that we wanted to be consistent with experimental validation, where we also used P@500. To check the difference between P@500 and AP@K, we computed AP@K for three top methods: MPS(B&T), MPS(T) and RNM with different K values (K=50, 100, 500, 2000). We found that the ranking of those three methods using AP@K (for all the K values we tested here) is consistent with that of using P@500 (see **Figure R4**). This result implies that P@500 is also a robust metric.

In the revised manuscript, we have included Fig.R4 as Fig.S5 in the SI, and added the following sentence in the main text (see page 7, lines 245-246):

“In particular, we found RNM and MPS(T) also shows high AP@K (average precision at K) against different K values (see Fig.S5).”

Point 2.2: 2. Figure S1, a-c, pearson's correlation is very misleading, spearman should be used.

Response: We apologize for this confusion. In Fig.S1a-c, we actually computed the Spearman's correlation, rather than Pearson's correlation. The “Pearson” correlation in the caption of Figure S1 was a typo, which has been fixed in the revised manuscript.

Point 2.3: 3. Figure 3a, it would be nice to leave a gap between different method groups. Also separate traditional similarity based methods from advanced similarity based methods as this is the major conclusion from the paper.

Response: We thank Review #2 for this very constructive comment. We have added the gap

between methods and separated the traditional similarity method and advanced similarity methods (see **Figure R2**).

Point 2.4: 4. The main text in Page 5 mentioned Fig.3a-e, but there are only two panels in Figure 3.

Response: We thank a lot for Review #2 pointing this out. We have revised the cross-reference accordingly.

Point 2.5: 5. Figure S2, the H. Sapiens network showed the lowest predictability but it is associated with the best performance in Figure 3, this needs to be explained.

Response: We thank Review #2 for this constructive comment. Indeed, the human interactome HuRI showed the lowest predictability, while various methods demonstrate high link-prediction performance (especially in terms of $P@500$) for this network. This seeming “inconsistency” is due to the fact that the predictability is essentially $P@10\%L$, i.e., Precision based on the removal of 10% of the total links, where L is the number of links in the network. For HuRI, $L=52,548$, so $10\%L\sim 5,255$. For this network, for any computational method $P@5,255$ will be much lower than $P@500$. To demonstrate this point, we computed $P@500$, $P@2000$ and $P@5000$ for three top methods: MPS(B&T), MPS(T), and RNM. The results are shown in **Figure R5**.

In the revised manuscript, we have included Fig.R5 as Fig.S3 in the SI. To avoid any confusion, in the revised main text, we have added the following sentences to explain this “inconsistency” (see page 5, lines 196-199):

“Note that the seeming “inconsistency” between the predictability and $P@500$ is because predictability is essentially $P@10\%L$. For HuRI this is about $P@5255$, which is much lower than $P@500$ (see Fig.S3).”

Point 2.6: 6. It would be nice to have a figure simultaneously visualizing average rankings of the 24 methods on the five networks and ranking variability

Response: We thank Reviewer #2 for this excellent suggestion. The average ranking and the ranking variability were visualized simultaneously in **Figure R6**. In the revised manuscript, we have included this figure as Fig.S4 in the SI.

Point 2.7: 7. The section “there are four consistently high-performing methods”, why cGAN is excluded? It looks good in Figure 3b.

Response: We thank Reviewer #2 for this comment. In the previous manuscript, we did not consider cGAN as a high-performance method because the high rank of cGAN is solely determined by one particular performance metric (i.e., NDCG). In terms of other three metrics, cGAN didn’t perform well. As the NDCG scores over different methods are quite similar, rendering a small standard deviation, the high NDCG of cGAN will lead to a very high Z-score.

In the revised manuscript, we evaluated the updated version of cGAN, as well as two additional methods: L3 and CRA. We also updated the performance of RepGSP. We have revised that section accordingly.

Point 2.8: 8. It seems that degree plays a major role in many algorithms. To investigate degree-driven prediction, it would be interesting to apply the prediction methods to networks generated by degree-preserving rewiring and check for their prediction performance.

Response: We thank Reviewer #2 for this very insightful comment. We have evaluated the performance of four methods: RNM, MPS(B&T), MPS(T) and SBM which tend to predict PPIs between high-degree proteins. We found that their performance dropped significantly on degree-preserving randomized interactomes (see **Figure R7**).

In the revised manuscript, we have included Fig.R7 as Fig.S8 in the SI, and added the following sentence in main text (see page 8, lines 300-302):

“The performance of all those four methods will degrade on degree-preserving randomized interactomes (see Fig.S8).”

Point 2.9: 9. Figure S8, it is strange to perform functional analysis of the predicted PPIs without putting them in the context of the known PPIs used for the prediction.

Response: We thank Reviewer #2 for this comment. In Figure S8, the reason why we did not put the predicted PPIs together with known PPIs is that we wanted to examine (in an unsupervised manner) whether those predicted PPIs are associated with particular functional cluster.

Point 2.10: 10. In experimental validation, is there a trend that predicted PPIs involving high-degree proteins are more likely to be validated?

Response: We thank Reviewer #2 for this comment. In the revised manuscript, we compared the degree of proteins for experimentally validated positive and negative PPIs. We found that those proteins involved in positive PPIs tend to have higher degrees than that involved in negative PPIs (see **Figure R8**).

In the revised manuscript, we have included Fig.R8 as Fig.S10 in the SI, and added the following sentence in the main text (see page 9, lines 351-352):

“We also found that those positive PPIs tends to connect high degree proteins (see Fig.S10).”

Finally, we thank Reviewer #2 again for her/his very constructive and insightful comments. We hope our responses have addressed those comments in a satisfactory manner.

Responses to Reviewer #3

Point 3.0: Summary.

Protein interactomes, that are often modeled as protein-protein interaction (PPI) networks, provide important insights into the functioning of the cell, which is key for understanding the molecular bases of biological processes and diseases. However, due to cost of experimentally capturing PPIs, interactomes remain largely unmapped. To overcome this limitation, many PPI prediction methods have been proposed in the literature.

In this paper, the authors present a systematic evaluation of the performances of network-based PPI prediction approaches, and also incorporate large-scale confirmatory experimentation (wet-lab).

- On real and synthetic interactomes, they evaluate the prediction performances of 24 state of the art methods. A key result is that the predictability of interactome is low, which may be due to the incompleteness of the PPI data. The authors also highlight four methods that consistently perform well, namely RNM, MPS(T), MPS(B&T), and SEAL.

- Then, the authors focus on the seven best performing approaches and further validate their top-500 PPI predictions on the human interactome using wet-lab experiments (using Y2H assay). From the corresponding 3,276 unique predictions, they validated 1,177 PPI between 633 proteins.

Assessment.

Overall, the paper is well written, and the proposed methodology to compare different PPI prediction methods is sound (it properly uses cross-fold validation and 4 different performance metrics), and the large scale wet-lab validation of the top predicted PPI is impressive. However, there are some issues.

Response: We thank Reviewer #3 for reviewing our manuscript and her/his positive assessment on our manuscript. Next, we address each of the reviewer comments in order.

Point 3.1: First, unlike claimed, this is not the first time that a large scale experimental validation is conducted to confirm the computational predictions of network based PPI prediction methods (e.g., see [1]). However, the scale of the wet-lab validation is impressive.

Response: We thank Reviewer #3 for this comment. We have revised that sentence as follows (see main text, page 9, lines 304-305):

“To the best of our knowledge, this is an unprecedentedly large-scale experimental validation of network-based PPI prediction methods in a systematical benchmark study.”

Point 3.2: Second, and more importantly, the computational comparative analysis is plagued by a poor choice of PPI data on which it is performed. As a consequence, the presented results and conclusions may not ‘generalize’ to the current wealth of PPI data available (more species, larger sizes, and larger variety of capturing technologies).

In particular, to avoid biases, the authors decided to cherry pick high quality PPI datasets. This leads to the following issues:

1- The computational comparison is based on only five (four real and one synthetic) interactomes, which is underwhelming and surprising given that the authors compared a large number of methods (24) and did a large scale wet-lab validation of the top-ranked PPI predictions. With only four real interactomes, the dataset is not representative of the large variety of available PPI data; e.g., there are far more interactome data available in databases such as BioGRID [2] or STRING [3]. For instance, in [1], the performance of L3-based PPI prediction method is evaluated on 16 interactomes from 7 species.

Response: We thank Reviewer #3 for this very constructive comment. In the revised manuscript, we have added the evaluations on two additional PPI datasets: BioGRID and STRING (see **Figure R2**). We found that all our previous conclusions (i.e., predictability of interactomes is weak, most methods vary considerably across different interactomes, traditional similarity-based methods do not perform well, etc.) still hold.

Point 3.3: 2- Also, the selected interactomes are very small / very incomplete, and may not represent the completeness of available PPI data (e.g., the human PPI data used in the study has 52K interactions between 8K proteins, while current human experimental PPI network from BioGRID database has 521K interactions between 19K proteins).

Response: We thank Reviewer #3 for this comment. In the revised manuscript, we also evaluated the prediction methods on the BioGRID database (see **Figure R2**). We found that those top-ranking methods, e.g., RNM, MPS(T), MPS(B&T) still show the superior performance in BioGRID.

Point 3.4: 3- Despite the claimed objective of reducing biases in the data, three out of the four real interactomes (C. elegans, yeast, and human) are captured using the same technology (Y2H), so a strong technological bias remains.

To overcome these limitations, the authors should include more interactomes, captured with a larger variety of bio-technologies, and with varying amount of experimental validation.

Response: We thank Reviewer #3 for this very constructive comment. In the revised manuscript, we have added the evaluations on two additional PPI databases: BioGRID and STRING (see **Figure R2, and Table S1**).

Point 3.5: As a minor comment, I don't see the benefit of using only one synthetic interactome. Often, random model network generators are used to generate 'controlled' datasets to study the effect of a given parameter (e.g., the number of nodes, the number of interaction edges, the amount of noise, or of the underlying network models), e.g., in the context of network comparison [4]. But with only one network, none of this is possible. For instance, the authors could have used synthetic networks generated with varying edge densities to further support their claim that more complete interactomes (i.e., with larger edge densities) are more predictable.

Response: We thank Reviewer #3 for this very insightful comment. In the revised manuscript, we removed the results and corresponding description of synthetic interactome. In addition, according to Reviewer #3's constructive comment, we computed the predictability of synthetic

interactomes with different edge densities. We found that dense interactomes are more predictable than sparse interactomes (see **Figure R9**).

In the revised manuscript, we have included Fig.R9 as Fig.S2 in the SI.

[1] Kovács et al. (2019). Network-based prediction of protein interactions. *Nature communications*, 10(1), 1-8.

[2] BioGRID database statistic: <https://wiki.thebiogrid.org/doku.php/statistics>

[3] STRING database statistics: https://string-db.org/cgi/about?sessionId=b9wz9kS1gbKk&footer_active_subpage=statistics

[4] Yaveroglu et al. (2015), Proper evaluation of alignment-free network comparison methods, *Bioinformatics*, 31(16):2697-2704.

Finally, we thank Reviewer #3 again for her/his very constructive and insightful comments. We hope our responses have addressed those comments in a satisfactory manner.

Figure R1: The difference between AUROC and AUPRC is due to data imbalance. (top) A highly imbalanced dataset with 100,000 negative instances and 50 positive instances. **(bottom)** A balanced dataset with 50 negative instances and 50 positive instances. Columns 1-3 shows the score distributions, the receiver operating characteristic (ROC) curve and the Precision-Recall curve (PRC) curve, respectively. Scores of positive instances were generated from a normal distribution with mean 0.15 and standard deviation 0.1; while scores of negative instances were generated from a normal distribution with mean 0 and standard deviation 0.1.

Figure R3: Functional analysis of proteins involved in the predicted PPIs. We firstly extracted the Gene Ontology (GO) terms for those proteins involved in the PPIs predicted by MPS(T) and MPS(B&T) using FuncAssociate¹. Then, we showed the number of proteins associated with each GO term. **a-b**, Top-20 GO terms associated with proteins involved in the top-500 PPIs predicted by MPS(T) (a) and MPS(B&T) (b). **c-d**, Top-20 GO terms associated with proteins involved in the top-5,000 PPIs predicted by MPS(T) (c) and MPS(B&T) (d).

Figure R4: AP@K and P@500 of three top methods: MPS(B&T), MPS(T) and RNM. AP@K ($K=50, 100, 500, 2000$) is computed as the average precision $P@i, i=1, \dots, K$. Error bar represents the standard deviation among ten realizations of HuRI.

Figure R5: P@K of three top methods: MPS(B&T), MPS(T) and RNM. Error bar represents the standard deviation among ten realizations of HuRI.

Figure R6: Ranking and variability of the PPI prediction methods over six interactomes. Bar represents the mean ranking of each method over six interactomes, and the error bar represents the variability of a method (computed as the standard derivation of rankings over six interactomes analyzed in this project). We did not show the variabilities of two methods that were not applied to all the six interactomes.

Figure R7: Performance of PPI prediction methods on the degree-preserved randomized interactome. We randomly rewired the PPIs of HuRI in $10N$ trials while preserving the original interactome's degree distribution using the functions *rewire* and *keeping_degreeseq* in the *igraph*² package. Four prediction methods RNM, SBM, MPS(T), and MPS(B&T) that tend to predict PPIs involving proteins of high degrees were evaluated in the randomized interactome. Error bar represents the standard deviation among 10-fold validations.

Figure R8: Degree distribution of proteins involved in experimentally validated PPIs. We compared the degree of proteins (in original HuRI) involved in experimentally validated positive and negative PPIs. P-value was calculated using Wilcoxon test.

Figure R9: Predictability of different interactomes. Boxplot shows the predictability over 50 different realizations. For each realization, we randomly split the links into the training set (90%), with the remaining 10% as the test set. To quantify the predictability of each interactome, we calculated its structural consistency index σ_c based on the first-order perturbation of the interactome's adjacency matrix, using the Matlab implementation of the Structural Perturbation Method (SPM) for link prediction [76]. (Note that here we explicitly considered self-loops in the calculations of σ_c .) See SI Sec.IA for details on SPM. Boxes indicate the interquartile range between the first and third quartiles with the central mark inside each box indicating the median. Whiskers extend to the lowest and highest values within 1.5 times the interquartile range. **a**, Predictability of real interactomes. **b**, Predictability of synthetic interactomes with different edge density. Here, we generated synthetic interactomes using duplication-mutation-complementation model³. Size of the synthetic interactome is 5,000 with a tuning divergence parameter.

Reference

1. Berriz, G. F., King, O. D., Bryant, B., Sander, C. & Roth, F. P. Characterizing gene sets with FuncAssociate. *Bioinformatics* **19**, 2502–2504 (2003).
2. Csardi, G. & Nepusz, T. The igraph software package for complex network research. *InterJournal, complex systems* **1695**, 1–9 (2006).
3. Vázquez, A., Flammini, A., Maritan, A. & Vespignani, A. Modeling of Protein Interaction Networks. *ComPlexUs* **1**, 38–44 (2003).

Reviewers' Comments:

Reviewer #1:

Remarks to the Author:

The authors have not sufficiently addressed the reviewer's doubts about the presented results. The changes to the manuscript are not sufficient enough to warrant a publication.

Specifically. Using an unbalanced set, as the authors do, is correct. We expect there is no interaction between the two random proteins in the full interactome. This means that AUROC is directly interpretable and we should expect not that distantly dissimilar performance for the presented method outside the benchmark. That AUROC of 0.9 on the balanced set could be very misleading on the real-life performance of the method, but on the unbalanced is closer to the real-world expectation. This leads to two possible interpretations of the given results: 1) The authors have a very biased benchmark 2) the authors almost completely solved the human interactome.

I do believe that the former statement explains the seen results which means that I do not believe that the following statements from the abstract hold:

Statement 1) "Our results indicate that advanced similarity-based methods, which leverage the underlying network characteristics of PPIs, show superior performance over other general link prediction methods."

Statement 2) "[...] finding 1,177 new 61 human PPIs (involving 633 proteins)."

Statement 3) "These results establish advanced similarity-based methods as powerful tools for the prediction of human PPIs."

Specifically:

Point 1.5:

The way the authors present the functional association (Fig. R3, Fig S13) is meaningless, as there is no testing involved. The authors just list the frequency of the GO terms. The term "binding" which is the first one is also associated with more than half of annotated human proteins. My guess is that is not statistically significant. Even more problematic is Fig. S12, which lists root terms of the respective Gene Ontology branches as being associated with the found clusters ("Molecular Function" and "Cellular Component")... to best of my knowledge these terms are associated with almost all proteins.

Point 1.6 and Point 1.7:

To my remark regarding the fact that top-scoring associations do not make much sense the authors respond, that yes, but they were not experimentally validated, so that's not a problem. But the authors' Statment 3) does not specify that you have to experimentally validate the "powerful tools to predict human PPI".

Point 1.8.

This statement that proteins share a GO term (organelle membrane) is not a statistical test. In fact, I've tested it, and no it's not a statistically significant term (as they are no statistically enriched terms for the set of proteins interacting with APOL2), so these proteins really do not have anything in common.

The authors do not prove that any of the newly discovered and validated human interactions is not just noise. With the exception of the set of keratins, but that is trivial and could result from the bias of the methods used. The performance testing is suboptimal, and small biases in the method can lead to misinterpretation of the AUC measures.

Reviewer #2:

Remarks to the Author:

The authors have adequately addressed my comments.

Reviewer #3:

Remarks to the Author:

My concerns were mostly addressed. The only remaining one is concerning the set of considered PPI networks. In the previous paper it was only 4 small PPI networks, and now it is 5 small and 1 large ppi networks. It would be good if this could be improved further.

Responses to Reviewer #1

Point 1.0: The authors have not sufficiently addressed the reviewer's doubts about the presented results. The changes to the manuscript are not sufficient enough to warrant a publication.

Response: We thank Reviewer #1 for reviewing our manuscript again. We apologize for not fully addressing her/his concerns on the presented results. Next, we address each of her/his remaining comments in order.

Point 1.1: Specifically. Using an unbalanced set, as the authors do, is correct. We expect there is no interaction between the two random proteins in the full interactome. This means that AUROC is directly interpretable and we should expect not that distantly dissimilar performance for the presented method outside the benchmark. That AUROC of 0.9 on the balanced set could be very misleading on the real-life performance of the method, but on the unbalanced is closer to the real-world expectation. This leads to two possible interpretations of the given results: 1) The authors have a very biased benchmark 2) the authors almost completely solved the human interactome.

Response: We thank Reviewer #1 for this critical comment.

The high AUROC values are due to the fact that the distribution of links is highly imbalanced in the PPI prediction problem. Evaluating PPI prediction methods using AUROC will overestimate their performance. We have described this point in our previous manuscript (please see Page 4, Lines 153-158):

“Considering that the distribution of links is highly imbalanced in the PPI prediction problem due to the sparsity of interactome maps across organisms^{24,25}, AUROC may overestimate the performance of a link prediction method, while AUPRC can provide more pertinent evaluation^{26,27}. Indeed, by systematically comparing the performance metrics of various PPI prediction methods, we found clear evidence that AUROC largely overestimates the performance of any particular method.”

The reason why we still presented the results on AUROC in our manuscript is simply because AUROC has been widely used in the link prediction literature [1-5], and we wanted to demonstrate clear evidence that it is not a good performance metric for the PPI prediction problem.

We emphasize that in this work AUROC was **not** included in the performance interpretation, comparison and ranking of different PPI prediction methods. Hence, it doesn't affect our assessment of different methods.

Point 1.2: I do believe that the former statement explains the seen results which means that I do not believe that the following statements from the abstract hold:

Statement 1) “Our results indicate that advanced similarity-based methods, which leverage the underlying network characteristics of PPIs, show superior performance over other general link prediction methods.”

Response: We thank Reviewer #1 for this critical comment. We apologize for not being sufficiently clear about the motivation of including results on AUROC. As mentioned in our

response to Point 1.1, AUROC will overestimate the performance of link prediction methods if the distribution of links is highly imbalanced. In this work, AUROC was not included in the performance interpretation, comparison and ranking of different prediction methods.

In the revised manuscript, to emphasize this point, we have revised the following sentence (see Page 5, Lines 175-176):

“Hereafter, we will, therefore, exclude AUROC in calculating the combined z-score **as well as interpretation of the performance** for each method.”

and added the following sentence to the caption of Figure 3 (see Page 27, Lines 974-975):

“**Note that AUROC was excluded in calculating the combined z-score and ranking for each method**”

Point 1.3: Statement 2 “[...] finding 1,177 new 61 human PPIs (involving 633 proteins).”

Response: We thank Reviewer #1 for this comment. Those 1,177 new human PPIs have been validated by the well-established yeast two-hybrid (Y2H) assays.

Point 1.4: Statement 3 “These results establish advanced similarity-based methods as powerful tools for the prediction of human PPIs.”

Response: We thank Reviewer #1 for this comment, and we apologize for not interpreting our results clearly. This statement does not rely on AUROC results. We have added some descriptions in the revised manuscript to avoid potentially misleading readers (please see our responses to Points 1.1 and 1.2).

Point 1.5: Specifically:

The way the authors present the functional association (Fig. R3, Fig S13) is meaningless, as there is no testing involved. The authors just list the frequency of the GO terms. The term “binding” which is the first one is also associated with more than half of annotated human proteins. My guess is that is not statistically significant.

Response: We thank Reviewer #1 for this critical comment. Those GO terms presented in Fig.R3 of our previous response letter (corresponding to Fig.S13 in the SI) were derived from the functional enrichment analysis tool --- FuncAssociate 3.0: The Gene Set Functionator [6], which has a default adjusted p-value threshold 0.05 (as other tools do). Hence, those GO terms are statistically significant. In **Table R1**, we listed all the 27 GOs (and their associated p-values) for proteins involved in those the top-5000 PPIs predicted by MPS(T). It is clear that the GO term “binding” and other GOs all have adjusted p-value < 0.05, and hence are statistically significant.

To further confirm the results shown in Table R1, we performed the functional enrichment analysis using another popular tool: g:Profiler [7] (which has been cited 2,479 times to date since its publication in 2019). The results are shown in **Table R2**. We found in total 123 GO terms (molecular function: 15; cellular component: 75; biological process: 33) that are statistically significant (with adjusted p-value < 0.05). Interestingly, 20 of the 27 GOs shown in Table R1 (highlighted in red) also appear in Table R2. These include the term: GO:0005488 (binding).

Table R1: Results of enriched GO terms for proteins evolved in the top-5000 PPIs predicted by MPS(T). The analysis was performed by using FuncAssociate3.0. The 20 GOs colored in red also appear in Table R2.

N	X	LOD	P	P_adj	attrib ID	attrib name
8	12	1.35492479	4.69E-07	0.004	GO:0005833	hemoglobin complex
8	14	1.19517645	2.47E-06	0.01	GO:0005344	oxygen transporter activity
64	122	1.13608048	8.40E-39	<0.001	GO:0045095	keratin filament
8	15	1.13300479	4.94E-06	0.018	GO:0015671	oxygen transport
113	220	1.13082364	1.44E-66	<0.001	GO:0031424	keratinization
10	19	1.12263506	3.59E-07	0.004	GO:0015669	gas transport
99	230	0.98130066	2.40E-49	<0.001	GO:0005882	intermediate filament
51	125	0.92912119	1.20E-24	<0.001	GO:0070268	cornification
13	42	0.74006178	1.02E-05	0.035	GO:0045104	intermediate filament cytoskeleton organization
13	43	0.72556017	1.36E-05	0.041	GO:0045103	intermediate filament-based process
15	54	0.67361837	9.54E-06	0.032	GO:0070936	protein K48-linked ubiquitination
78	609	0.25578491	6.86E-06	0.02	GO:0005198	structural molecule activity
138	1113	0.2459277	1.48E-08	0.001	GO:0005615	extracellular space
174	1493	0.21772705	1.72E-08	0.001	GO:0044430	cytoskeletal part
1065	11770	0.20722112	1.09E-17	<0.001	GO:0005515	protein binding
344	3249	0.18571551	9.10E-11	<0.001	GO:0044707	single-multicellular organism process
348	3308	0.18249248	1.51E-10	<0.001	GO:0032501	multicellular organismal process
390	3847	0.16586632	1.00E-09	<0.001	GO:0044421	extracellular region part
296	2935	0.15251379	3.44E-07	0.004	GO:0070062	extracellular exosome
296	2936	0.15232154	3.55E-07	0.004	GO:0065010	extracellular membrane-bounded organelle
296	2954	0.14887126	6.17E-07	0.005	GO:1903561	extracellular vesicle
296	2955	0.14868015	6.36E-07	0.005	GO:0043230	extracellular organelle
330	3347	0.14301666	5.48E-07	0.005	GO:0031988	membrane-bounded vesicle
346	3536	0.1402022	5.60E-07	0.005	GO:0031982	vesicle
429	4563	0.12433034	1.42E-06	0.008	GO:0044767	single-organism developmental process
1216	14846	0.11728911	1.28E-05	0.038	GO:0005488	binding
455	4965	0.11070852	1.03E-05	0.035	GO:0032502	developmental process

Table R2: Results of enriched GO terms for proteins evolved in the top-5000 PPIs predicted by MPS(T). The analysis was performed by using g:Profiler. The 20 GOs colored in red also appear in Table R1.

source	term_name	term_id	adjusted_p_value	term_size
GO:MF	protein binding	GO:0005515	3.78E-179	14832
GO:MF	binding	GO:0005488	1.10E-53	18303
GO:MF	structural constituent of skin epidermis	GO:0030280	2.31E-17	37
GO:MF	identical protein binding	GO:0042802	4.4942E-06	2121
GO:MF	haptoglobin binding	GO:0031720	1.9346E-05	10
GO:MF	signaling receptor regulator activity	GO:0030545	0.00073118	543
GO:MF	oxygen carrier activity	GO:0005344	0.00196675	15
GO:MF	signaling receptor activator activity	GO:0030546	0.00202526	508
GO:MF	receptor ligand activity	GO:0048018	0.00232018	500
GO:MF	ubiquitin-like protein conjugating enzyme activity	GO:0061650	0.00478199	37
GO:MF	ubiquitin conjugating enzyme activity	GO:0061631	0.01157567	34
GO:MF	molecular carrier activity	GO:0140104	0.01445746	82
GO:MF	growth factor activity	GO:0008083	0.01860679	165
GO:MF	signaling receptor binding	GO:0005102	0.02797529	1555
GO:MF	cAMP-dependent protein kinase regulator activity	GO:0008603	0.02949999	11
GO:BP	keratinization	GO:0031424	1.75E-28	83
GO:BP	intermediate filament organization	GO:0045109	2.50E-28	68
GO:BP	intermediate filament cytoskeleton organization	GO:0045104	5.22E-27	88
GO:BP	intermediate filament-based process	GO:0045103	9.91E-27	89
GO:BP	keratinocyte differentiation	GO:0030216	6.20E-18	169
GO:BP	skin development	GO:0043588	5.78E-14	305
GO:BP	epithelial cell differentiation	GO:0030855	8.83E-14	711
GO:BP	epidermal cell differentiation	GO:0009913	1.48E-13	239
GO:BP	epithelium development	GO:0060429	2.54E-12	1208
GO:BP	epidermis development	GO:0008544	5.40E-12	368
GO:BP	tissue development	GO:0009888	5.84E-12	1973
GO:BP	regulation of cellular process	GO:0050794	1.82E-11	11130
GO:BP	cellular developmental process	GO:0048869	1.41E-10	4280
GO:BP	cell differentiation	GO:0030154	1.59E-10	4256
GO:BP	developmental process	GO:0032502	2.66E-10	6424
GO:BP	anatomical structure development	GO:0048856	7.41E-10	5836
GO:BP	animal organ development	GO:0048513	2.05E-09	3591
GO:BP	response to stimulus	GO:0050896	2.75E-08	9000
GO:BP	regulation of response to stimulus	GO:0048583	7.91E-08	3970
GO:BP	carbon dioxide transport	GO:0015670	2.19E-07	15

GO:BP	biological regulation	GO:0065007	3.27E-07	13096
GO:BP	negative regulation of cellular process	GO:0048523	3.90E-07	4749
GO:BP	multicellular organismal process	GO:0032501	6.48E-07	7463
GO:BP	regulation of primary metabolic process	GO:0080090	8.92E-07	5869
GO:BP	regulation of nitrogen compound metabolic process	GO:0051171	2.5351E-06	5706
GO:BP	regulation of biological process	GO:0050789	5.8397E-06	12361
GO:BP	cellular response to stimulus	GO:0051716	1.1209E-05	7494
GO:BP	positive regulation of biological process	GO:0048518	1.7013E-05	6309
GO:BP	gas transport	GO:0015669	1.7016E-05	24
GO:BP	one-carbon compound transport	GO:0019755	3.0808E-05	25
GO:BP	response to stress	GO:0006950	3.6801E-05	3938
GO:BP	regulation of cellular metabolic process	GO:0031323	4.656E-05	5649
GO:BP	positive regulation of cellular process	GO:0048522	7.6728E-05	5641
GO:BP	cellular response to chemical stimulus	GO:0070887	0.0002596	3049
GO:BP	cell communication	GO:0007154	0.00037831	6551
GO:BP	regulation of signal transduction	GO:0009966	0.00040327	2968
GO:BP	positive regulation of response to stimulus	GO:0048584	0.00046351	2214
GO:BP	supramolecular fiber organization	GO:0097435	0.00054132	807
GO:BP	response to organic substance	GO:0010033	0.00055217	3034
GO:BP	signal transduction	GO:0007165	0.00055857	5993
GO:BP	cellular response to organic substance	GO:0071310	0.00079057	2405
GO:BP	signaling	GO:0023052	0.00130489	6492
GO:BP	regulation of axon regeneration	GO:0048679	0.00158057	28
GO:BP	response to external stimulus	GO:0009605	0.00178341	2814
GO:BP	regulation of signaling	GO:0023051	0.00290519	3374
GO:BP	regulation of cell communication	GO:0010646	0.00313583	3363
GO:BP	regulation of transcription by RNA polymerase II	GO:0006357	0.00319086	2574
GO:BP	regulation of cell death	GO:0010941	0.0032573	1637
GO:BP	transcription by RNA polymerase II	GO:0006366	0.00378545	2676
GO:BP	cell death	GO:0008219	0.00398029	2107
GO:BP	regulation of neuron projection regeneration	GO:0070570	0.00521021	31
GO:BP	defense response to other organism	GO:0098542	0.00575601	1163
GO:BP	regulation of molecular function	GO:0065009	0.00631256	3055
GO:BP	secretion by cell	GO:0032940	0.00702023	808
GO:BP	negative regulation of biological process	GO:0048519	0.008205	5918
GO:BP	protein K48-linked ubiquitination	GO:0070936	0.0104892	67
GO:BP	oxygen transport	GO:0015671	0.01130858	17
GO:BP	regulation of nucleobase-containing compound metabolic process	GO:0019219	0.01238068	4088

GO:BP	cell population proliferation	GO:0008283	0.01456825	1988
GO:BP	defense response	GO:0006952	0.01888476	1742
GO:BP	locomotion	GO:0040011	0.01995959	1349
GO:BP	positive regulation of signal transduction	GO:0009967	0.02300705	1523
GO:BP	regulation of RNA metabolic process	GO:0051252	0.02410591	3775
GO:BP	negative regulation of response to stimulus	GO:0048585	0.02444799	1604
GO:BP	negative regulation of multicellular organismal process	GO:0051241	0.02556781	1035
GO:BP	regulation of DNA-templated transcription	GO:0006355	0.03163657	3474
GO:BP	regulation of nucleic acid-templated transcription	GO:1903506	0.03235427	3475
GO:BP	export from cell	GO:0140352	0.03364887	866
GO:BP	cell surface receptor signaling pathway	GO:0007166	0.03462974	2816
GO:BP	DNA-templated transcription	GO:0006351	0.03713049	3595
GO:BP	nucleic acid-templated transcription	GO:0097659	0.03795358	3596
GO:BP	regulation of RNA biosynthetic process	GO:2001141	0.03954254	3484
GO:BP	RNA biosynthetic process	GO:0032774	0.04209486	3615
GO:BP	regulation of biosynthetic process	GO:0009889	0.04251432	4175
GO:BP	regulation of response to external stimulus	GO:0032101	0.04533914	948
GO:CC	intermediate filament	GO:0005882	1.08E-58	217
GO:CC	intermediate filament cytoskeleton	GO:0045111	2.06E-52	255
GO:CC	keratin filament	GO:0045095	1.36E-49	102
GO:CC	polymeric cytoskeletal fiber	GO:0099513	3.68E-28	779
GO:CC	supramolecular polymer	GO:0099081	4.57E-25	1026
GO:CC	supramolecular fiber	GO:0099512	4.15E-24	1017
GO:CC	extracellular space	GO:0005615	8.48E-22	3368
GO:CC	extracellular region	GO:0005576	1.04E-20	4302
GO:CC	supramolecular complex	GO:0099080	1.11E-20	1377
GO:CC	cytoplasm	GO:0005737	6.78E-18	12270
GO:CC	vesicle	GO:0031982	2.47E-14	3973
GO:CC	extracellular exosome	GO:0070062	9.35E-14	2108
GO:CC	extracellular vesicle	GO:1903561	1.84E-13	2132
GO:CC	extracellular membrane-bounded organelle	GO:0065010	1.94E-13	2133
GO:CC	extracellular organelle	GO:0043230	1.94E-13	2133
GO:CC	cytosol	GO:0005829	1.00E-11	5419
GO:CC	endoplasmic reticulum	GO:0005783	1.4465E-06	2031
GO:CC	cytoskeleton	GO:0005856	5.3323E-06	2381
GO:CC	haptoglobin-hemoglobin complex	GO:0031838	2.027E-05	11
GO:CC	hemoglobin complex	GO:0005833	0.00013997	13
GO:CC	endomembrane system	GO:0012505	0.00031646	4733

GO:CC	extracellular matrix	GO:0031012	0.00035086	565
GO:CC	external encapsulating structure	GO:0030312	0.00037412	566
GO:CC	collagen-containing extracellular matrix	GO:0062023	0.00102482	429
GO:CC	cytoplasmic vesicle	GO:0031410	0.00363285	2487
GO:CC	intracellular vesicle	GO:0097708	0.00383297	2489
GO:CC	endoplasmic reticulum membrane	GO:0005789	0.00858259	1162
GO:CC	endoplasmic reticulum subcompartment	GO:0098827	0.00995922	1166
GO:CC	nuclear outer membrane-endoplasmic reticulum membrane network	GO:0042175	0.01220072	1184
GO:CC	endoplasmic reticulum protein-containing complex	GO:0140534	0.01295723	127
GO:CC	endoplasmic reticulum lumen	GO:0005788	0.02244166	312
GO:CC	organelle membrane	GO:0031090	0.03647855	3685
GO:CC	cAMP-dependent protein kinase complex	GO:0005952	0.04977835	9

Point 1.6: Even more problematic is Fig. S12, which lists root terms of the respective Gene Ontology branches as being associated with the found clusters (“Molecular Function” and “Cellular Component”)... to best of my knowledge these terms are associated with almost all proteins.

Response: We thank Reviewer #1 for this critical comment. The GOs in Figure.S12 were obtained from running FuncAssociate3.0 [6]. The functional modules were discovered by SAFE (spatial analysis of functional enrichment) [8]. This figure was used to demonstrate the functional relationships/modules among the new human PPIs discovered in this study.

Point 1.7: To my remark regarding the fact that top-scoring associations do not make much sense the authors respond, that yes, but they were not experimentally validated, so that’s not a problem. But the authors’ Statement 3) does not specify that you have to experimentally validate the “powerful tools to predict human PPI”.

Response: We thank Reviewer #1 for this comment. Our statement was based on both computational and experimental validations. Those advanced similarity-based methods, which leverage the underlying network characteristics of PPIs, show superior performance over other general link prediction methods in both computational validation among all six interactomes (see Figure 3) and experimental validation (see Figure 5). Note that those statements do not rely on AUROC results.

Point 1.8: This statement that proteins share a GO term (organelle membrane) is not a statistical test. In fact, I’ve tested it, and no it’s not a statistically significant term (as they are no statistically enriched terms for the set of proteins interacting with APOL2), so these proteins really do not have anything in common.

Response: We thank Reviewer #1 for this comment.

The statement that those six genes: APOL2, ELOVL4, ERGIC3, FAM210B, SLC35C2, and TIMMDC1 share a GO term (organelle membrane) was based on the functional enrichment analysis using g:Profiler, rather than FuncAssociate3.0. We apologize for not making this point clear in our previous response letter.

As shown in **Fig.R1**, both the GO term (organelle membrane) and the GO term (endoplasmic reticulum-Golgi intermediate compartment membrane) have adjusted p-value less than 0.05, and hence are statistically significant.

GO:CC	stats								
Term name	Term ID	Padj	$-\log_{10}(P_{adj})$	ENSG00000128335	ENSG00000118402	ENSG00000125991	ENSG00000124098	ENSG00000080189	ENSG00000113845
organelle membrane	GO:0031090	4.326×10^{-3}	3.5	Green	Red	Orange	Red	Blue	Green
endoplasmic reticulum-Golgi intermediate compartmen...	GO:0033116	3.337×10^{-2}	1.5	Yellow					

1 to 2 of 2 | Page 1 of 1

Figure R1: Functional enrichment analysis of a set of proteins interacting with APOL2. The analysis was performed by using g:Profiler.

Point 1.9: The authors do not prove that any of the newly discovered and validated human interactions is not just noise. With the exception of the set of keratins, but that is trivial and could result from the bias of the methods used. The performance testing is suboptimal, and small biases in the method can lead to misinterpretation of the AUC measures.

Response: We thank Reviewer #1 for this comment.

As shown in Fig.R3 of our previous response letter, those proteins involved in the predicted PPIs do show many statistically significant GO terms.

Regarding the AUROC, as we mentioned in our response to Point 1.1, we were aware of that AUROC will overestimate the performance of a link prediction method for imbalanced dataset, thus AUROC were not considered in ranking methods tested in this work. The reason that we calculated the AUROC is because (1) it has been widely used in link prediction papers [1-5]; (2) we wanted to explicitly demonstrate that it is quite misleading in evaluating methods of predicting PPIs.

As many of those predicted PPIs have been experimentally validated, this indicates that they are not just noise.

Finally, we thank Reviewer #1 again for reviewing our manuscript. We hope our responses above have fully addressed her/his concerns.

Responses to Reviewer #2

The authors have adequately addressed my comments.

We thank Reviewer #2 for reviewing our manuscript again. We are glad to know that the reviewer is satisfied with our previous response.

Responses to Reviewer #3

Point 3.0: My concerns were mostly addressed.

Response: We thank Reviewer #3 for reviewing our manuscript again.

Point 3.1: The only remaining one is concerning the set of considered PPI networks. In the previous paper it was only 4 small PPI networks, and now it is 5 small and 1 large ppi networks. It would be good if this could be improved further.

Response: We thank Reviewer #3 for this comment. We apologize for not better explaining our motivations of focusing on those interactomes used in this study.

We know that, to evaluate the performance of PPI prediction methods, we need reliable and unbiased benchmark interactomes. Literature-curated interactomes of PPIs with multiple lines of supporting evidence might be highly reliable, but they are largely influenced by selection biases. Therefore, in this study we focused on interactomes emerging from systematic screens that lack selection biases. For simplicity, we mainly focused on binary datasets where co-complex membership annotations are not included. Hence, we focused on the following benchmark interactomes for performance evaluation: (1) A plant interactome including 2,774 proteins and 6,205 PPIs, derived from the PPIs in the *A. thaliana* Interactome, version 1 (AI-1) and literature databases; (2) a worm interactome including 2,528 proteins and 3,864 PPIs, derived from *C. elegans* version 8 (WI8), which is assembled from high-quality Y2H PPIs; (3) a yeast interactome of *S. cerevisiae* including 2,018 proteins and 2,930 PPIs, derived from the union of CCSB-Y11, Ito-core and Uetz-screen datasets; (4) a human interactome including 8,274 proteins and 52,548 PPIs, derived from HuRI, which is assembled from binary protein interactions from three separate high-quality Y2H binding assays. Note that all the interactomes were downloaded from the interactome database (<https://ccsb.dana-farber.org/interactome-data.html>) maintained by the Center for Cancer Systems Biology (CCSB) at Dana-Farber Cancer Institute (DFCI).

Following the excellent suggestion made by Reviewer #3 (“*there are far more interactome data available in databases such as BioGRID [2] or STRING [3]*”), we have also tested two additional human interactomes in BioGRID and STRING. Together with the four benchmark interactomes mentioned above, we now have in total 6 interactomes in the current version of our manuscript (see **Fig.R2**).

Interactomes					
A. thaliana	C. elegans	S. cerevisiae	H. sapiens (HuRI)	H. sapiens (STRING)	H. sapiens (BioGRID)
2,774 proteins	2,528 proteins	2,018 proteins	8,274 proteins	6,926 proteins	19,665 proteins
6,205 PPIs	3,864 PPIs	2,930 PPIs	52,548 PPIs	41,948 PPIs	713,793 PPIs

Figure R2: Benchmark interactomes considered in this study.

We believe that this set of six interactomes (from four species) is comprehensive enough to assess various PPI prediction methods. Moreover, our computational validation demonstrated that there are five consistently high-performing methods, i.e., RNM, MPS(T), MPS(B&T), RepGSP and SEAL, which yield relatively high AUPRC and P@500 in their computational evaluations over the six interactomes.

Finally, we thank Reviewer #3 again for reviewing our manuscript. We hope our responses above have fully addressed her/his concerns.

References

- [1] Fawcett, T. An introduction to ROC analysis. *Pattern recognition letters* 27, 861–874 (2006).
- [2] Davis, J. & Goadrich, M. The relationship between Precision-Recall and ROC curves. in *Proceedings of the 23rd international conference on Machine learning* 233–240 (2006).
- [3] Yang, Y., Lichtenwalter, R. N. & Chawla, N. V. Evaluating link prediction methods. *Knowledge and Information Systems* 45, 751–782 (2015).
- [4] Clauset, A., Moore, C. & Newman, M. E. Hierarchical structure and the prediction of missing links in networks. *Nature* 453, 98–101 (2008).
- [5] Liu, W. & Lü, L. Link prediction based on local random walk. *EPL (Europhysics Letters)* 89, 58007 (2010).
- [6] Berriz, GF, et al. Characterizing gene sets with funcassociate, *Bioinformatics* 19, 2502 (2003).
- [7] Raudvere, U. et al. g:Profiler: a web server for functional enrichment analysis and conversions of gene lists (2019 update), *Nucleic Acids Research* 47(W1), W191–W198. (2019).
- [8] A. Baryshnikova, Systematic functional annotation and visualization of biological networks, *Cell Systems* 2, 412 (2016).

Reviewers' Comments:

Reviewer #1:

Remarks to the Author:

The authors have successfully addressed my remaining concerns.

Reviewer #3:

Remarks to the Author:

As already mentioned in my previous review of the article, most of my concerns have been addressed.

The only remaining one is concerning the fact that the performances and behaviors of the tested PPI are only assessed on a small number of PPI networks. After asking twice for using more data, the paper is still using only 5 small and one large PPI networks (including three different versions of the Human PPI network that may largely overlap).

While I understand the difficulty of testing 26 PPI prediction methods on a larger number of PPI network, the used PPI network dataset is too small to be considered as representative of the large quantity, variety and quality of the available PPI data. Thus, the claims made by the authors should be toned down throughout the paper.

Importantly, this limitation should be indicated in the Discussion section of the paper.

Response to Reviewer #3

Point 3.0: As already mentioned in my previous review of the article, most of my concerns have been addressed.

We thank Reviewer #3 very much for reviewing our manuscript again.

Point 3.1: The only remaining one is concerning the fact that the performances and behaviors of the tested PPI are only assessed on a small number of PPI networks. After asking twice for using more data, the paper is still using only 5 small and one large PPI networks (including three different versions of the Human PPI network that may largely overlap).

Response: We thank Reviewer #3 for this comment. In the first version of our manuscript, we evaluated 24 prediction methods on four Y2H-based PPI datasets and one synthetic dataset. In the revised version, we tested two additional human PPI datasets using 26 methods. There are several reasons why we did not test those methods on many other existing PPI datasets:

- First, we emphasize that, to evaluate the performance of various PPI prediction methods, we need reliable and unbiased benchmark interactomes. Literature-curated interactomes of PPIs with multiple lines of supporting evidence might be highly reliable, but they are largely influenced by selection biases. Therefore, in this work we focused on interactomes emerging from systematic screens that lack selection biases.
- Second, in a previous work¹, Kovács et al. have already evaluated the performance of the L3-based PPI prediction method on 16 interactomes from 7 species. They found that L3 displays superior performance in those interactomes (e.g., interactomes of *S. cerevisiae*, *A. thaliana*, *C. elegans*). Those L3-based methods, e.g., L3, RNM and MPS, still show superior performance than other methods in the interactomes considered in the current study.
- Third, we tested two more human interactomes (rather than more interactomes of other species), because the experimental validation focused on human PPIs.
- Last but not least, it's computationally very challenging to systematically validate all the 26 methods over too many interactomes.

We apologize for not making those points more explicit to the reviewer in our previous responses. We hope the reviewer could understand our choice now.

Point 3.2: While I understand the difficulty of testing 26 PPI prediction methods on a larger number of PPI network, the used PPI network dataset is too small to be considered as representative of the large quantity, variety and quality of the available PPI data. Thus, the claims made by the authors should be toned down throughout the paper.

Response: We thank Reviewer #3 for this comment. In the revised manuscript, we have toned down our conclusions throughout the paper, including

- the Abstract (see page 2, line 58): "...advanced similarity-based methods, which leverage the underlying network characteristics of PPIs, show superior performance over other general link prediction methods **in the interactomes we considered.**"

- the Introduction section (see page 4, line 118): “We found that advanced similarity-based methods, which leverage the underlying characteristics of PPIs, show superior performance over other link prediction methods in both computational and experimental validations **in the interactomes we considered.**”
- and the Discussion section (see page 11, line 397): “By contrast, link prediction methods MPS and RNM, which leverage specific connectivity properties of PPI networks (i.e., the L3 principle), displayed the most promising performance **in the interactomes we considered.**”

Point 3.3: **Importantly, this limitation should be indicated in the Discussion section of the paper.**

Response: We thank Reviewer #3 for this excellent suggestion. In the revised manuscript, we have added this point in the Discussion section (see page 12, lines 439-441):

“In addition, we only considered six interactomes from four species, which certainly does not cover the variety and quality of all the available PPI datasets from different species.”

Finally, we thank Reviewer #3 again for reviewing our manuscript. We hope our responses above have fully addressed her/his concerns.

Reference

1. Kovács, I. A. *et al.* Network-based prediction of protein interactions. *Nat Commun* **10**, 1240 (2019).